# Benefits of Steroid Injections into Paraspinous Muscles After Spinal Surgery in a Rat Paraspinal Muscle Retraction Model

**DOI:** 10.3390/ijms262211093

**Published:** 2025-11-16

**Authors:** Meei-Ling Sheu, Liang-Yi Pan, Jason Sheehan, De-Wei Lai, Yu-Cheng Chou, Liang-Yu Pan, Chien-Chia Wang, Ying Ju Chen, Hong-Lin Su, Hsi-Kai Tsou, Hung-Chuan Pan

**Affiliations:** 1Institute of Biomedical Science, National Chung Hsing University, Taichung 402, Taiwan; mlsheu@nchu.edu.tw (M.-L.S.); pan0911606850@gmail.com (L.-Y.P.); 2Doctoral Program in Biotechnology Industrial Management and Innovation, National Chung Hsing University, Taichung 402, Taiwan; 3Ph.D. Program in Translational Medicine, Rong Hsing Research Center for Translational Medicine, National Chung Hsing University, Taichung 402, Taiwan; suhonhlin@gmail.com; 4Department of Medical Research, Taichung Veterans General Hospital, Taichung 407, Taiwan; deweilai123@vghtc.gov.tw; 5Department of Neurosurgery, University of Virginia, Charlottesville, VA 22908, USA; jps2f@virginia.edu (J.S.); chouycns@vghtc.gov.tw (Y.-C.C.); 6Faculty of Medicine, Poznan University of Medical Sciences, 61-701 Poznan, Poland; mattpan9009@gmail.com; 7Department of Life Sciences, National Central University, Taoyuan 320, Taiwan; superdukewang@gmail.com; 8Ph.D. Program in Health and Social Welfare for Indigenous Peoples, Providence University, Taichung 433, Taiwan; yichen5@pu.edu.tw; 9Department of Neurosurgery, Neurological Institute, Taichung Veterans General Hospital, Taichung 407, Taiwan; hktsou@vghtc.gov.tw

**Keywords:** paraspinous muscle, steroid, lumbar surgery, muscle regeneration

## Abstract

Open posterior lumbar surgery involves detaching paraspinal muscles from the spine to decompress neural tissues and to place instruments. While this operation improves the quality of life, it often has adverse effects on skeletal muscles like inflammation, degeneration, and fibrosis. Corticosteroids are well known for their anti-inflammatory function. In this study, we assessed the protective effects of intramuscular injection of corticosteroid on injured paraspinal muscles following surgery on the spine. C2C12 cells were co-exposed to hypoxia and lipopolysaccharide (LPS) to simulate ischemia and inflammatory response after muscle retraction to assess the effect of steroid. In vivo experiment, animals first underwent paraspinous muscle splitting with retractors to induce muscle injury, and later were assessed for neurobehavior, electrophysiology, and protein level related to inflammatory or regeneration following intramuscular (IM) steroid injection. Steroid rescued reduced cell viability caused by hypoxia + LPS, and attenuated induced protein expression of iNOS, COX2, Bad, and Bax. In neurobehavioral assessments (CatWalk, Ethovision, Von Frey test, and open field locomotor), retraction of paraspinous muscles worsened behaviors that were improved by IM steroid injections. The electrophysiology study showed that IM steroid injection lessened the muscle denervation caused by retraction. Similarly, IM steroid injections also attenuated dorsal root ganglion antigenicity of CGRP, Iba-1, and CD68 induced by muscle retraction. Muscle retraction downregulated AChR, desmin, PSD 95, and GAP 43, whereas IM steroid injection attenuated the adverse effects. The restoration of muscle morphology and decreased fibrosis were also facilitated by IM dexamethasone. IM steroid injection appears to protect against retraction damage in paraspinous muscle following spinal surgery. IM steroid paraspinous muscle injection may provide beneficial effects in spinal operations.

## 1. Introduction

Surgery-related morbidity remains a crucial concern in efforts to optimize postoperative outcomes and enhance patients’ quality of life. Among major spinal procedures, open posterior approaches are frequently employed but are inherently associated with iatrogenic injury to the multifidus and erector spinae muscles—an issue of continuing concern to spine surgeons. These procedures require detachment of the paraspinal musculature to expose neural elements and facilitate decompression or instrumentation. Although open posterior lumbar surgery can significantly improve function and quality of life in most patients, it may also produce adverse musculoskeletal effects that compromise long-term recovery [1,2,3,4,5].

The paraspinal muscles, which act as key extensors to maintain sagittal balance, are particularly vulnerable to damage during posterior spinal exposure. Reported postoperative atrophy of these muscles ranges from 10% to 40% of their preoperative volume [6,7,8,9]. Kawaguchi and colleagues demonstrated that prolonged retraction time and excessive pressure from self-retaining retractors elevate intramuscular pressure and impair perfusion, leading to ischemic necrosis—factors recognized as major contributors to surgical muscle injury [10]. Similarly, Hu et al. observed in a rabbit model that retraction of the multifidus muscle caused acute edema, necrosis, and inflammation within the first postoperative week, with worsening severity as retraction duration increased [11]. Histological observations revealed early regeneration during the first week, fibrotic remodeling between the third and sixth weeks, and progressive fatty degeneration between the twelfth and twenty-fourth weeks. Longer retraction times produced higher fibrosis and fat-infiltration scores, indicating that even minimal manipulation may result in irreversible changes such as atrophy, fibrosis, and fatty degeneration [6,7,8,9,12,13].

Experimental and clinical evidence has identified multiple intraoperative contributors to multifidus injury, with muscle splitting and retraction recognized as primary factors [14]. Investigations have examined these injuries from several perspectives, including pathogenesis [15,16,17,18], cellular and molecular responses [19,20], histopathological alterations [10,17,21], imaging manifestations [22,23], functional impairment [22,24], and clinical sequelae [10,17,22,24,25]. Intermittent retraction and similar modifications have been proposed to mitigate these detrimental effects [25].

The pathophysiological cascade of paraspinal muscle damage during surgery closely resembles ischemia–reperfusion injury. Temporary ischemia followed by reperfusion triggers excessive production of reactive oxygen and nitrogen species (ROS/RONS) and activates inflammatory cascades [26,27,28]. The resultant oxidative stress promotes expression of pro-inflammatory cytokines, chemokines, and adhesion molecules, leading to infiltration by neutrophils and macrophages. These immune cells perpetuate the cycle of ROS generation and cytokine release, thereby amplifying inflammation and tissue injury. Moreover, activated macrophages release transforming growth factor-β1 (TGF-β1), a potent profibrotic mediator that drives collagen deposition and scar formation [26].

Following skeletal muscle injury, a stereotypical sequence ensues involving degeneration, inflammation, and regeneration [29]. Myofiber rupture and necrosis disrupt the extracellular matrix and vascular network, attracting inflammatory cells. Neutrophils arrive first, followed by monocytes that differentiate into macrophages within 24–48 h, clearing necrotic debris through phagocytosis [30]. Together with fibroblasts and extracellular matrix components, macrophages release cytokines, chemokines, and growth factors that orchestrate repair [31,32,33]. After skeletal muscle injury, a coordinated sequence of degeneration, inflammation, and regeneration occurs. Inflammation plays a dual role—transient inflammatory activity is essential for clearing necrotic debris and stimulating regeneration, whereas persistent or excessive inflammation promotes fibrosis and pain, ultimately impairing skeletal muscle function. However, persistent inflammation promotes fibrosis [34] and pain, ultimately impairing muscle function [35]. Hence, clinical strategies aimed at minimizing surgical trauma and controlling inflammation are essential to preserve paraspinal muscle integrity and optimize postoperative recovery.

Corticosteroids are anti-inflammatory agents because they inhibit the infiltration of monocytes and neutrophils to the site of inflammation. Corticosteroids also block T-cell activation by inhibiting cytokine release, leading to lower levels of interleukins and TNF-α in the tissue [36]. To date, evidence supports that corticosteroids may be beneficial in the short term, but with chronic use, cause injury to healing muscles, like disrupting fiber integrity, incomplete healing, and reductions in force-generating capacity [37,38]. In a small series study on lumbar discectomy, incisional infiltration of paravertebral muscles preemptively with bupivacaine and methylprednisolone provided pain control similar to bupivacaine alone [39]. On the other hand, in a large series study, infiltration of methylprednisolone as an adjunct to ropivacaine before wound closure was reported to be a safe and efficient strategy for pain management following laminoplasty or laminectomy [40]. In brief, steroid injections into the paraspinal muscle appear to have protected muscle against injury after spinal surgery.

Conventional analgesic strategies—such as systemic opioids, nonsteroidal anti-inflammatory drugs (NSAIDs), and muscle relaxants—are often limited by their short duration of efficacy, systemic adverse effects, and poor local anti-inflammatory control at the surgical site. Opioids, while effective in acute pain relief, are associated with tolerance, dependence, and respiratory depression. Similarly, NSAIDs may compromise bone healing and increase gastrointestinal and renal complications. These limitations underscore the need for alternative or adjunctive therapies that can locally attenuate inflammation and promote functional recovery. Local corticosteroid administration, by contrast, provides targeted suppression of inflammatory cascades—reducing cytokine release, leukocyte infiltration, and oxidative damage within traumatized paraspinal muscles. Therefore, this study explores the therapeutic efficacy of local steroid injection in attenuating retraction-induced muscle injury, inflammation, and fibrosis.

In this study, we adopted two approaches. First, C2C12 cells under the condition of hypoxia + LPS were used to mimic the muscle retraction with dual blood vessels compromised and an inflammation reaction to investigate the effect of steroids in protection against muscle injury. Second, with an animal model, we first applied the paraspinous muscle retraction for an hour with a self-retaining retractor. We then infiltrated the paraspinous muscle with steroids to assess the muscle protection and the suppression of the inflammatory response.

## 2. Results

### 2.1. Reduced Cell Viability in Hypoxia + LPS Condition Rescued by Dexamethasone Treatment

The MTT assay was used to assess the survival of C2C12 cells as a function of dexamethasone doses, under normoxic or hypoxic + LPS conditions. We found that cell survival dropped as the dosage increased up to 500 μM under normoxic conditions (Figure 1A,C). Under the hypoxic conditions co-administrated with LPS, a decrease in cell viability was observed. However, with increasing doses of dexamethasone in hypoxic + LPS conditions, an increase in cellular viability was noted, starting at 10 μM and continuing in a dose-dependent manner up to 250 μM. At the dose of 500 μM, adverse effects were observed (Figure 1B,D). The MTT assay was used to assess the survival of C2C12 cells as a function of dexamethasone dose under either normoxic or hypoxic + LPS conditions. Under normoxic conditions, increasing dexamethasone concentrations up to 500 μM resulted in a progressive decline in cell viability, suggesting that excessive glucocorticoid exposure is cytotoxic to C2C12 myoblasts (Figure 1A,C). In contrast, under hypoxic conditions combined with LPS stimulation, which mimics an inflammatory and metabolic stress environment, a marked reduction in cell survival was observed compared with normoxia.

However, dexamethasone treatment under hypoxia + LPS conditions produced a dose-dependent rescue of cell viability starting at 10 μM, reaching maximal protection around 250 μM. This recovery likely reflects the anti-inflammatory and metabolic-stabilizing effects of glucocorticoids, which can suppress LPS-induced signaling and modulate stress-responsive pathways such as NF-κB and HIF-1α. At 500 μM, however, the protective effect was lost, indicating that supra-physiological concentrations may instead induce cytotoxic or metabolic stress responses (Figure 1B,D).

Collectively, these findings demonstrate a biphasic response to dexamethasone, with moderate doses restoring cell viability under inflammatory hypoxia while high doses become detrimental, emphasizing the need for dose optimization in future translational applications.

### 2.2. Alleviation of Neuroinflammation and Apoptosis in C2C12 Cells by the Administration of Dexamethasone

Under hypoxic + LPS conditions, a pronounced upregulation of the inflammatory mediators iNOS and COX-2 was observed, indicating activation of pro-inflammatory signaling pathways (Figure 2A,B). This inflammatory response was progressively attenuated by dexamethasone in a dose-dependent manner up to 100 μM, beyond which no additional suppression was achieved at 250 μM. The plateau suggests that the anti-inflammatory effect of dexamethasone reaches a saturation point, likely due to maximal glucocorticoid receptor activation at this concentration.

Similarly, pro-apoptotic proteins Bad and Bax were markedly elevated under hypoxic + LPS stress, consistent with mitochondrial-mediated apoptosis triggered by inflammatory and metabolic insults. Dexamethasone treatment reduced the expression of these apoptotic markers beginning at 50 μM and reaching maximal inhibition around 100 μM, mirroring the anti-inflammatory trend. Conversely, the anti-apoptotic protein Bcl-2 was downregulated by hypoxia + LPS but restored by dexamethasone in a dose-dependent manner, reflecting the compound’s ability to stabilize mitochondrial integrity and promote cell survival. These reciprocal changes between pro- and anti-apoptotic proteins confirm that dexamethasone not only mitigates inflammation but also prevents secondary apoptotic damage under stress conditions (Figure 2A,B). Immunocytochemical analysis revealed expression patterns of Bax, Bad, and Bcl-2 that closely paralleled the Western blot data, providing consistent morphological validation of the biochemical findings (Figure 2C).

Collectively, these results demonstrate that dexamethasone exerts dual protective effects—suppressing inflammatory enzyme expression and attenuating apoptosis—in C2C12 myoblasts subjected to inflammatory hypoxia.

### 2.3. Modulation of Inflammatory Response in Dorsal Root Ganglion Cells Under Different Treatment Regimens

A paraspinous muscle retraction model was established to evaluate neuroinflammatory responses under mechanical stress, and animals were divided into four groups: sham, muscle retraction, muscle retraction + intramuscular (IM) dexamethasone, and muscle retraction + intravenous (IV) dexamethasone. Dorsal root ganglia (DRG) were harvested and analyzed by immunohistochemistry for CGRP, Iba-1, and CD68, representing markers of nociceptive activity and microglial/macrophage activation, respectively. A significant increase in CGRP expression was observed in the DRG of the muscle retraction group compared with the sham control, indicating enhanced nociceptive signaling following mechanical stress (Figure 3A,D). This elevation was markedly reduced by IM dexamethasone administration, suggesting effective local suppression of inflammation-induced neuropeptide release. In contrast, IV dexamethasone produced only a modest reduction, implying that systemic administration may result in insufficient local concentration to fully suppress the peripheral inflammatory cascade. Similarly, expression of the microglial activation marker Iba-1 increased markedly after muscle retraction, reflecting enhanced neuroinflammatory activation in the DRG (Figure 3B,D).

IM dexamethasone significantly reduced Iba-1 immunoreactivity, whereas IV administration had minimal effect, further supporting that targeted local delivery achieves superior anti-inflammatory efficacy at the neural interface. The macrophage marker CD68 exhibited a parallel distribution pattern to Iba-1 (Figure 3C,D), reinforcing the notion that both resident and infiltrating immune cells contribute to post-retraction inflammation.

Together, these findings indicate that local (IM) dexamethasone administration more effectively attenuates nociceptive and microglial activation in DRG than systemic (IV) delivery, highlighting the importance of tissue-specific drug distribution in modulating neuroinflammation after surgical muscle retraction.

### 2.4. Improvement in Neurobehavior Following Paraspinous Muscle Retraction with Intramuscular Injection of Dexamethasone

In CatWalk gait analysis, static (print length, print intensity) and basic spatiotemporal parameters (stance, swing, maximum contact intensity, single-stance) did not differ among groups, indicating that gross motor output and weight-bearing capacity were largely preserved. By contrast, the regularity index (RI)—a composite measure of inter-limb coordination—declined significantly after muscle retraction relative to sham, consistent with a coordination-specific deficit rather than generalized motor impairment. Intramuscular (IM) dexamethasone restored the RI toward sham levels, whereas intravenous (IV) dexamethasone produced no significant improvement, suggesting that adequate local tissue exposure is required to normalize coordination after peripheral inflammatory insult (Figure 4A).

In the EthoVision assay, muscle retraction reduced the ratio of novel exploration, a behavioral readout often linked to pain-related avoidance and anxiety-like states. IM dexamethasone increased novel exploration toward control values, whereas IV dexamethasone did not, supporting a preferential effect of local delivery on behaviors sensitive to nociceptive burden (Figure 4B).

Thermal nociception (hot-plate) was comparable across groups, implying that heat-evoked pathways were not prominently affected in this paradigm. In contrast, mechanical withdrawal thresholds were significantly decreased by muscle retraction (mechanical allodynia) and were restored by IM—but not IV—dexamethasone, pointing to a dominant component of peripheral mechanical hypersensitivity that is mitigated by targeted anti-inflammatory treatment (Figure 4C).

In the open-field test, muscle retraction reduced total distance traveled and time in the center zone, consistent with hypo-locomotion and anxiety-like behavior under ongoing nociceptive stress. Both measures were normalized by IM dexamethasone, whereas IV dosing showed no significant effect, again highlighting the importance of local pharmacokinetics for behavioral rescue (Figure 4D,E).

Taken together, these multimodal behavioral data indicate that muscle retraction induces coordination deficits, mechanical hypersensitivity, and anxiety-like avoidance that are selectively reversed by IM dexamethasone. The dissociation between preserved gross motor parameters and impaired RI/mechanical thresholds supports a model in which local neuroinflammation drives circuit-level coordination and sensory abnormalities rather than frank motor weakness. The superior efficacy of IM over IV dosing aligns with our DRG immunohistochemistry (Section 2.3), reinforcing the rationale for local anti-inflammatory strategies to achieve functional recovery.

### 2.5. Increased Muscle Regeneration and Restoration of Muscle Morphology in Paraspinous Muscle Following Intramuscular Injection of Dexamethasone

Paraspinous muscles were harvested 7 days after retraction injury and analyzed using immunohistochemistry, Western blot, and ELISA. Immunohistochemical staining revealed a marked accumulation of CD68-positive macrophages after muscle retraction, reflecting robust local inflammation. This CD68 elevation was significantly reduced by intramuscular (IM) dexamethasone, while intravenous (IV) administration produced only minimal attenuation, a finding further validated by Western blot analysis (Figure 5A,F,G). These results indicate that local dexamethasone delivery effectively suppresses macrophage-driven inflammation at the injury site.

ELISA quantification showed that muscle retraction induced significant increases in pro-inflammatory cytokines IL-1β, IL-6, and TNF-α, all of which were suppressed by IM dexamethasone but not by IV administration (Table 1).

This pattern highlights that local glucocorticoid availability is essential for modulating the early inflammatory cytokine milieu critical to subsequent tissue repair.

Morphologically, immunohistochemistry demonstrated severe disruption of muscle fiber organization following retraction, accompanied by reduced staining intensity for AChR, PSD-95, and Desmin, indicating impaired neuromuscular junction integrity and cytoskeletal damage. IM dexamethasone markedly restored both structural integrity and marker expression intensity, whereas IV administration resulted in only subtle improvement (Figure 5B–D,F,G). These results suggest that local anti-inflammatory treatment not only mitigates inflammation but also preserves neuromuscular connectivity and cytoskeletal stability essential for regeneration.

To further assess regenerative capacity, GAP-43—a marker of axonal sprouting and myogenic repair—was examined. GAP-43 expression decreased sharply after muscle retraction, but was substantially upregulated by IM dexamethasone, while IV treatment had no effect (Figure 5E–G). This indicates that localized glucocorticoid therapy promotes both neural and myogenic regeneration pathways following mechanical injury.

At 28 days post-operation, EMG and histological analyses were performed to evaluate long-term muscle recovery. H&E staining revealed a marked reduction in muscle bulk and fiber density in the retraction group, which was largely restored in the IM dexamethasone group, while IV administration again yielded only modest recovery (Figure 6A, Table 2). Sirius Red staining confirmed that retracted muscles were replaced by extensive fibrotic tissue, whereas IM dexamethasone markedly reduced collagen deposition, indicating prevention of pathological fibrosis (Figure 6B,C, Table 2). Electromyographic recordings demonstrated frequent fibrillation potentials in the muscle retraction group—reflecting ongoing denervation—which were significantly attenuated by IM dexamethasone, with only minor improvement in the IV group (Figure 6D).

Together, these findings reveal that local IM dexamethasone administration exerts superior anti-inflammatory, anti-fibrotic, and pro-regenerative effects compared to systemic IV delivery. By effectively reducing macrophage infiltration, cytokine expression, and fibrosis while restoring muscle architecture, neuromuscular integrity, and electrophysiological function, IM dexamethasone emerges as a promising therapeutic strategy for improving structural and functional recovery after paraspinous muscle injury.

## 3. Discussion

In this study, we have provided a simple platform by intramuscular injection of steroids to prevent muscle atrophy and post-operative pain in open spinal surgery. When myocytes had been exposed concomitantly to hypoxia + LPS, mimicking the effects of muscle retraction, the resulting apoptosis was reduced by steroid treatment. In the animal study, we found that the effects of IM steroids were better than those of systemic administration of steroids. The benefits of IM steroids over IV ones help to avoid the systemic complications associated with IV steroid injections. In this study, the intramuscular injection of steroids provided a simple and effective method to prevent muscle atrophy and post-operative pain in open spine surgery.

The protective effect of steroids in muscle injury is an ongoing controversy. Chronic myopathy occurred after using corticosteroids (prednisolone) for >4 weeks at a dose > 10 mg/day. Acute myopathy, on the other hand, is more commonly associated with critically ill patients receiving doses > 60 mg/day [41]. Interestingly, the frequency of steroid administration also plays a critical role. For example, Klfl5, a key transcription factor targeted by steroids, increases weekly treatments, but the Klfl5 levels decrease with daily treatment [42]. The loss of Klf15 exacerbates skeletal muscle phenotypes in mdx mice, a model for muscular dystrophy [43]. Also, daily steroids administration activates atrophic pathways, including F-box 32 (Fbxo32), which encodes atrogin-1, a protein linked to muscle atrophy. In contrast, weekly steroid administration in mdx mice improves muscle function and histopathology, while simultaneously inducing the ergogenic transcription factor Klf15 and reducing Fbxo32 expression. These findings suggest that intermittent steroid regimens, rather than daily administration, promote sarcolemma repair and muscle recovery while minimizing atrophic remodeling [44]. Moreover, a single intramuscular injection of a combination of bupivacaine and methylprednisolone provides superior pain control in spinal surgery [39,40]. Similarly, trigger point steroid injection therapy demonstrates significant early recovery benefits in patients suffering from chronic lumbosacral radiculopathy with trigger points [45,46]. Hence, the protective effects of steroids on muscle injury appear to be closely tied to the timing and dosing of administration, with short-term or intermittent regimens showing the greatest promise. In this study, the observed benefits of a single intramuscular steroid injection are in line with the principle of short-term and intermittent administration, reinforcing its efficacy in promoting muscle recovery while mitigating adverse effects.

Macrophages are known to lyse target muscle cells through a nitric oxide (NO)-dependent but superoxide-independent mechanism [47]. The presence of muscle cells further enhances macrophage-derived NO production, suggesting a possible positive-feedback loop that amplifies tissue damage [47]. Thus, initial muscle injury may potentiate NO-mediated cytotoxicity by macrophages. In addition, ischemia–reperfusion injury provoked a robust inflammatory response characterized by cytokine release, which in turn induced Inos expression [48]. These findings collectively underscore the pivotal role of macrophages in promoting muscle damage following injuries under conditions of mechanical stress and ischemia. The temporal pattern of COX-2 expression after muscle injury parallels the inflammatory response, and pharmacological inhibition of COX-2 has been shown to alleviate post-inflammatory fibrosis [19]. In our present study, C2C12 myoblasts co-treated with hypoxia and LPS exhibited significant upregulation of iNOS and COX-2, both of which were markedly attenuated by dexamethasone treatment. Furthermore, paraspinous muscle retraction in vivo resulted in pronounced macrophage accumulation, which was similarly reduced following IM steroid injections. Together, these in vitro and in vivo findings suggested that IM steroid administration promotes muscle recovery, primarily by suppressing macrophage activity and dampening the inflammatory cascades.

To simulate the pathophysiology process of paraspinous muscle retraction during spinal surgery, we employed an in vitro model in which C2C12 myocytes were co-exposed to hypoxia and LPS. This approach reasonably reproduces the ischemic and inflammatory milieu seen in surgical settings. Hypoxia is known to regulate satellite cell activation, self-renewal, proliferation, and differentiation at sites of skeletal muscle injury [49,50]. LPS, acting through TLR4, activates the ubiquitin–proteasome pathway, leading to enhanced protein catabolism and muscle cell degradation in cultured C2C12 muscle cells [51]. Furthermore, LPS impairs myogenic differentiation, thereby contributing to skeletal muscle atrophy [52], and promotes the expression of pro-inflammatory cytokines not only in immune tissues but also in skeletal muscle [53]. The effect of dexamethasone on myocytes is context-dependent, varying with differentiation stage, dosage, and frequency of administration; thus, its impact can be either deleterious or protective [4,44,54]. Muscle apoptosis, a form of programmed cell death, can arise during or following ischemic episodes. Ischemia induces cellular stress through calcium overload and reactive oxygen species (ROS) production, both of which can activate apoptotic pathways [54,55]. In this study, we observed that combined hypoxia and LPS treatment inhibited myocyte differentiation in a dose-dependent manner, while high concentrations (up to 500 μM) produced overt cytotoxic effects. Importantly, dexamethasone treatment mitigated these apoptotic responses, suggesting a cytoprotective effect under hypoxic and inflammatory stress. These results further support the therapeutic rationale for intramuscular steroid administration during spinal surgery, as it may attenuate inflammation-induced muscle apoptosis and promote postoperative muscle recovery.

Dorsal root ganglion (DRG) cells are located within the intervertebral foramen of the spinal cord, and they are sensory neurons that respond to peripheral nerve injury [56,57]. Calcitonin gene-related peptide (CGRP) plays a critical role in the transmission and modulation of pain signals [58,59]. Approximately 45% of DRG neurons express CGRP, predominantly those with small-diameter unmyelinated C-fibers [60]. Animal models of adjuvant arthritis and chronic inflammatory pain reported significant upregulations of CGRP and its mRNA in DRG neurons [58,61]. Several cytokines, including IL-1β, IL-6, and TNF-α, have been linked to DRG neuron excitability by sensitizing TRP channels, which, in turn, stimulate CGRP release [62,63,64,65,66]. Following peripheral nerve damage, macrophages invade the DRG, where cell bodies of damaged nerve fibers are located [67,68]. Peripheral nerve injuries also lead to activation of microglia in the DRG [69,70]. This process is paralleled by a significant expansion and proliferation of macrophages around injured DGR sensory neurons [71]. In our present study, muscle injury resulted in a significant expression of CGRP, which was attenuated by IM steroid injections. This attenuation paralleled the invasions of macrophages and microglia into DRG. Additionally, muscle retraction led to significant increases in a number of inflammatory cytokines, like TNF-α, INF-1β, and IL-6. These increments were significantly attenuated by IM steroid injections. Results indicated that the IM injections of steroid effectively alleviated the inflammatory response. This alleviation contributed to a reduction in CGRP expression and suppressed microglia and macrophage invasion. These changes were consistent with weaker pain-related neurobehavior.

Muscle retraction during surgery increases intramuscular pressure and decreases blood flow, leading to edema, necrosis, and inflammation [10,11,17,26,27,28]. This pathological cascade activates polymorphonuclear cells, monocytes, and macrophages, which, along with fibroblasts and the extracellular matrix, release growth factors, cytokines, and chemokines [30,31,32,33]. Inflammation contributes to fibrosis [34] and generates pain, which can impair skeletal muscle function [35]. In our present study, we found that IM steroid injections attenuated macrophage activation-thereby reducing the associated inflammatory cytokines. This effect was paralleled by a drop in muscle fibrosis.

The restoration of muscle function after injury involves neurotrophic factors that speed up the axonal flow speed [72]. GAP-43, expressed in skeletal muscle fibers, plays a critical role in muscle regeneration, with greater expression observed during this process [73,74]. Several studies have reported varying degrees of fragmentation and acetylcholine receptor dispersion in denervated muscles [73,75], PSD-95 clusters receptors, ion channels, signaling molecules, and cytoskeletal components near the postsynaptic neuron surface to facilitate synaptic transmission. These expressions are downregulated following muscle injury [73,76]. Desmin, a major intermediate filament protein essential for the structural integrity and function of muscle, is reported to link with muscle functional recovery [73,77,78]. In our present study, muscle retraction significantly lowered expression of GAP-43, acetylcholine receptors, PSD-95, and Desmin. These markers were restored by IM steroid injection, further confirming the regeneration of injured muscle.

Paraspinal muscle retraction causes varying degrees of pain behavior, including neuropathic pain, and may induce psychological symptoms [79,80]. The regularity index (RI) parameter in CatWalk gait analysis primarily measures inter-limb coordination, which is strongly correlated with neuropathic pain and memory deterioration [56,81,82,83]. Memory impairment is demonstrated in assessments using EthoVision XT [81,84]. The open-field test is widely used to study the neurobiological basis of anxiety and to screen for novel drug targets or anxiolytic compounds [85]. Also, the mechanical withdrawal threshold aligns with the severity of neuropathic pain [56,84]. In our study, paraspinal muscle retraction resulted in neuropathic pain, as evidenced by lower RI and mechanical withdrawal thresholds. Paraspinal muscle retraction also led to memory impairment and depression-like behaviors, such as reduced novel object exploration and shorter walking distances. These neurobehavioral impairments were significantly alleviated by IM steroid injections.

The translational relevance of the steroid dose used in rats requires careful interpretation for clinical application. Although this dose effectively reduced inflammation and fibrosis, direct extrapolation to humans must consider interspecies differences in body surface area, metabolic rate, and tissue pharmacokinetics. Allometric scaling can guide dose estimation, but variations in local perfusion and muscle volume may significantly alter corticosteroid bioavailability and therapeutic duration. Site specificity and delivery also present key translational challenges. The local distribution of steroids depends on injection depth, diffusion distance, and muscle vascularization, all of which differ between rodent and human paraspinal musculature. Optimized delivery systems or sustained-release formulations may therefore be necessary to achieve comparable tissue exposure in clinical settings.

Finally, limitations of the current rat retraction model must be acknowledged. The model reproduces acute, single-level ischemic injury but does not replicate the chronic, multi-level retraction or prolonged paraspinal damage characteristic of complex human spinal fusion. These disparities in surgical duration, tissue stress, and mechanical loading may influence fibrosis and recovery, underscoring the need for large-animal or chronic compression models to bridge preclinical and clinical translation.

In our present study, we aimed to address a knowledge gap in existing literature, namely, on the use of steroid injections in paraspinous muscles for pain relief and preservation of muscle injury following spinal surgery in animals. To simulate the conditions of paraspinous muscle retraction, we combined hypoxia with LPS to mimic ischemia and inflammatory cell infiltration. The C2C12 monoculture system was employed to examine the direct, cell-autonomous response of myoblasts to inflammatory stimuli. While this model does not recapitulate the full in vivo inflammatory milieu involving immune–muscle cross-talk, it provides insight into how myoblasts respond to bacterial components such as LPS via TLR4-dependent signaling. Both in vitro and in vivo experiments demonstrated that a single dose of steroid significantly improved myocyte survival and alleviated pain-related behavior. Therefore, IM steroid injections after spinal surgery appeared to be a reasonable approach to minimize muscle injury after spinal surgery.

## 4. Materials and Methods

### 4.1. Cell Culture

C2C12 murine skeletal muscle myoblasts (passages 15–30, ECACC, Salisbury, UK) were cultured in tissue culture flasks (Sarstedt AG & Co. KG, Nümbrecht, Germany), in a humidified, 37 °C, 5% CO_2_ incubator and passaged every 2 to 3 days at 80–90% confluency [73]. Growth media (GM) for cell propagation was DMEM (4.5 g/L glucose, containing GlutaMAX (Thermo Fisher Scientific, Gibco™, Waltham, MA, USA) added with sterile supplements according to the following composition (per 100 mL): 10 mL FCS (gamma-irradiated, containing 1.16 g/L glucose; BioSera, Nuaille, France), 1 mL sodium pyruvate (100 mM), 0.5 mL Penicillin (10,000 U/mL), streptomycin (10,000 ug/mL), and 1 mL uridine. To simulate a hypoxic condition, cells were transferred into a humidified incubator (Innova CO-48) (Eppendorf/New Brunswick Scientific, Edison, NJ, USA) set to maintain a 1% O_2_, 5% CO_2_, 37 °C environment. C2C12 cells under the condition of hypoxia + LPS (100 μg) for 24 h [51] were used to mimic the muscle retraction with dual blood vessels compromised and an accompanying inflammation reaction, thereby allowing us to investigate the effects of steroids in protection against muscle injury.

### 4.2. MTT Assays

The MTT assay was performed to evaluate cellular metabolic activity, serving as an indirect measure of cell viability, proliferation, and cytotoxicity. This colorimetric method relies on the enzymatic reduction of the yellow tetrazolium compound, 3-(4,5-dimethylthiazol-2-yl)-2,5-diphenyltetrazolium bromide (MTT), into insoluble purple formazan crystals by metabolically active cells. The reaction is catalyzed by NAD(P)H-dependent oxidoreductase enzymes present in viable cells, reflecting mitochondrial function and overall metabolic integrity. Following incubation, the formazan crystals were dissolved using a solubilization reagent, and the optical density of the resulting colored solution was measured at a wavelength between 500 and 600 nm with a multi-well spectrophotometer. The intensity of the color produced was directly proportional to the number of living, metabolically active cells, with darker absorbance indicating higher cell viability [86].

### 4.3. Western Blot

Protein expressions in both paraspinous muscle and treated C2C12 cells were determined by Western blotting. In brief, proteins (60 μg) were separated by SDS-PAGE, electrophoretically transferred to nitrocellulose membranes, and blocked for 1 h in phosphate-buffered saline (PBS) containing Tween 20 (0.1%) and non-fat milk (5%). Blots were incubated for 1 h with the following iNOS (ABN26#, 1:000, Millipore, Burlington, MA, USA), COX2 (BD-6102-3#, 1:000, BD, San Jose, CA, USA), Bad (Sc-8044#, 1:1000, Santa, Dallas, TX, USA), Bax (ab32503#, 1:1000, Abcam, Cambridge, UK), Bcl-2 (610539#, 1000, BD, San Jose, CA, USA), AChR (ab308306#, 1:1000, Abcam), GAP 43 (GTX127937#, 1:000, GeneTex, Irvine, CA, USA), PSD 95 (GTX133091#, 1:1000, GeneTex, Irvine, CA, USA), Desmin (ab32362#, 1:1000, Abcam, Cambridge, UK), CD 68 (MAB 1435#, 1:000, Millipore, Burlington, MA, USA), and GAPDH (sc-32233#, 1:1000, Santa, Dallas, TX, USA). Membranes were finally incubated for 1 h with a horseradish peroxidase-conjugated secondary antibody. After further washing with PBS, blots were incubated with commercial chemiluminescence reagents (Amersham Biosciences, Amersham, UK).

### 4.4. Immunohistochemistry

After treatment, C2C12 cells were fixed for the immunohistochemistry staining. In brief, treated C2C12 cells were incubated in 0.2% Triton X-100 for 30 min, rinsed twice in PBS with 0.5% bovine serum albumin (BSA), and then incubated overnight at room temperature with the appropriate primary antibodies. Primary antibodies used were against one of the following: Bad (Sc-8044#, 1:1000, Santa), Bax (ab32503#, 1:1000, Abcam), and Bcl-2 (610539#, 1000, BD). L4–L6 dorsal root ganglia were bilaterally harvested from the different experimental groups and subjected to immunohistochemical staining. Dorsal root ganglion samples obtained on day 7 were analyzed using immunohistochemistry staining for the following markers: CGRP (ab81887#, 1:250, Abcam,), Iba1 (NB 100-1028#, 1:500, Novus, Centennial, CO, USA), and CD 68 (MAB 1435#, 1:500, Millipore). The paraspinous muscle samples detached from the spinous processes and lamina were cryosectioned into 8-μm-thick sections cut longitudinally along the spinal axis and mounted on Superfrost Plus slides (Menzel-Gläser, Braunschweig, Germany). Paraspinous muscle samples harvested on day 7 underwent immunohistochemistry for CD 68 (MAB 1435#, 1:500, Millipore), AChR (ab308306#, 1:1000, Abcam), PSD 95 (GTX133091#, 1:1000, GeneTex), Desmin (ab32362#, 1:1000, Abcam), and GAP 43 (GTX127937#, 1:000, GeneTex). Treated C2C12 myoblasts, dorsal root ganglion (DRG) neurons, and paraspinal muscle tissue sections were first rinsed with phosphate-buffered saline (PBS) and then incubated for 1 h at room temperature with fluorescently labeled secondary antibodies. The following antibody combinations were used: FITC-conjugated anti-mouse IgG (1:200; Kirkegaard & Perry Laboratories, Gaithersburg, MD, USA) with Texas Red-conjugated anti-rabbit IgG (1:200; Vector Laboratories, Stuttgart, Germany), or Texas Red-conjugated anti-mouse IgG (1:200; Vector Laboratories) with FITC-conjugated anti-goat IgG (1:200; Organon Teknika, Durham, NC, USA). After incubation, samples were washed and mounted on glass slides using Vectashield mounting medium (Vector Laboratories).

Fluorescent images were acquired using an Olympus IX71 confocal laser scanning microscope (Olympus, Shinjuku, Japan). Quantitative analysis of immunofluorescence intensity was performed with ImageJ software (Version 1.52, NIH, Bethesda, MD, USA). Briefly, the immunostained images were imported and separated into three individual color channels. Unnecessary channels were closed, and the relevant channel was saved as a TIFF file. Threshold adjustment was applied to isolate positively stained regions against a dark background. Binary images were then generated, converting all stained (positive) pixels to black and non-stained (negative) pixels to white. The number of black pixels represented the area of positive immunoreactivity, expressed as black pixels per mm^2^. Quantitative measurements were obtained from the dermal region of paw skin and the L4–L5 DRG (*n* = 6 per group), and results were reported as fluorescence density or pixel counts for each analyzed tissue section [73].

### 4.5. Animal Model

Sprague–Dawley rats weighing 250 to 300 g (from National Laboratory Animal Center, Taipei, Taiwan) were used in this study with approval from Institutional Animal Care and Use Committee or the Panel of Taichung Veterans General Hospital (La-1132063) and in accordance with ARRIVE guidelines. Rats were first induced under anesthesia with 4% isoflurane, followed by a maintenance dose of 1 to 2%. Surgical procedures involved paraspinous muscle splitting with a retractor to induce paraspinal muscle ischemia for a duration of 1 h (Figure 7). Rats were divided into 4 groups: sham, muscle retraction (MR), MR + intramuscular (IM) dexamethasone (1 mg), and MR + intravenous (IV) dexamethasone (1 mg). The dexamethasone dosage (1 mg) was calculated based on the human-equivalent dose adjusted for a 250 g Sprague–Dawley rat [87]. One milligram of dexamethasone was diluted in phosphate-buffered saline (PBS) to a final volume of 100 μL. The solution was administered bilaterally into the paraspinal muscles at a rate of 10 μL/min using a 26-gauge Hamilton syringe. The needle was maintained in position for an additional 5 min after injection to minimize backflow and ensure proper local diffusion.

The following neurobehavioral assessments were conducted: Catwalk gait analysis, Ethovision, Von Frey test, and open field evaluation. Baseline measurements were taken 3 days before injury, followed by weekly assessments for the 4 weeks post-surgery until the end of the study. At the end of the experiment, the animals were first anesthetized with 4% isoflurane and then transcardially perfused with 4% paraformaldehyde in 0.1 M phosphate buffer (pH 7.4). Dorsal root ganglion cells and paraspinous muscles were collected for immunohistochemistry, Western blot, and ELISA analyses.

### 4.6. Thermal Hyperalgesia and Mechanical Allodynia

For the mechanical allodynia test, each rat was placed on a customized platform fixed inside a transparent acrylic chamber (measuring 20 × 20 × 20 cm^3^). The platform itself measured 20 × 20 cm^2^ and was constructed from 5 mm thick acrylic plate, featuring a grid of 2 mm diameter holes spaced 5 mm apart in perpendicular rows across the entire surface. Each trial involved applying a von Frey hair (Touch-Test Sensory Evaluator, North Coast Medical, Inc., Gilroy, CA, USA) to the hind paw, repeated 5 times at 5-s intervals, or immediately after the hind paw was appropriately positioned on the platform. If the hind paw withdrawal failed to occur during 5 repeated trials, the hair was replaced by a hair of larger size. The hind paw withdrawal threshold of a hair size (represented in grams) was determined when withdrawal had occurred in 4 or more out of 5 trials.

Thermal hyperalgesia was assessed using the classical hot-plate test (Technical & Scientific Equipment GmbH, TSE systems, Bad Homburg, Germany). The paw withdrawal latency, defined as the time from the paw touching the 52 °C hot plate to the paw withdrawal, was recorded using a timer. A maximum cut-off time of 20 s was adopted to prevent tissue damage [56].

### 4.7. CatWalk-Automated Quantitative Gait Analysis

The CatWalk XT system was equipped with a high-speed digital camera that captured video at a sample rate of 100 frames/s. This camera transforms each scene (the area in front of the lens) into a digital image composed of discrete pixels with varying brightness values. These digital images are transferred to a computer via an Ethernet connection. The brightness of each pixel depends on the amount of light in the area captured by the camera. The Illuminated FootprintTM enabled detection of intensity differences between animals’ paws. The 3D Footprint Intensity Tab plots the print intensity of the four paws for each frame where the paws were in contact with the glass plate, and results are displayed in a 3D chart. Intensity values ranged from 0 to 225 and were color-coded. The 3D chart was able to be rotated in all directions.

Quantitative analysis in the CatWalk XT included the following parameters:

Step Sequence Distribution: 6 different walking patterns or normal step sequences were identified in rats, and were classified based on the sequential placements of paws into three categories.

Regularity Index (RI): This parameter measures inter-limb coordination. Coordination is considered normal when rats use only regular step sequences during uninterrupted locomotion. RI is calculated as: RI = (NSSP × 4/PP) × 100%, where NSSP is the number of normal step sequence patterns and PP is the total number of paw placements. Extra paw placement, or irregular walking on three paws, reduces RI.

Print Area: This parameter measures the total floor area in pixels contacted by a paw during the stance phase. An increase in hind limb print area may indicate lower limb paralysis or deficiency in plantar stepping or paw/toe dragging during part of the step cycle. A drop in this parameter suggests mechanical allodynia.

Base of Support: This is the distance in mm between the two hind paws, measured perpendicular to the direction of walking.

Duration of Swing and Stance Phases: These parameters depend on the animal’s walking speed and degree of dysfunction. They are expressed as fractions of the total step duration using the formula: fraction of stance or swing phase = (time in stance or swing phase/time in single step) × 100%. Durations are measured in seconds.

Hind Paw Pressures: This parameter represents the mean intensity of the hind paw’s contact area at the moment of maximal paw-floor contact, expressed in arbitrary units (a.u.) [56].

### 4.8. Electromyography

One month after muscle injury, needle electromyography (EMG) was performed on each animal shortly before euthanasia. After anesthetizing the rat, a concentric EMG needle (25 × 0.3 mm) was inserted into the paraspinous muscle to record abnormal spontaneous electromyographic activity. The ground electrode was a surface electrode attached to the contralateral hind paw. The electromyography examination consisted of 3 stages. In the 1st stage, the potential recorded by the electrode upon insertion into the muscle is referred to as insertion potential activity. In the 2nd stage, electrical activity at rest was recorded by advancing the electrode to the various depths of the muscle. These two stages provided the relevant information about muscle activity at rest. The 3rd stage involved motor evoked potential testing. Here, the spine was manipulated to induce motor action potential. Only fibrillation potentials and positive sharp were considered as abnormal spontaneous activity [73].

### 4.9. ELISA

We measured IL-1β (Catalog#: DY401-05: R&D, Minneapolis, MN, USA), IL-6 (Catalog#: M6000B-1: R&D), and TNF-α (Catalog#: TA00B-1; R&D) using commercially available kits according to the manufacturer’s instructions. Equal amounts of protein obtained from the paraspinous muscle were used for ELISA.

### 4.10. EthoVision XT with Novel Object Test

The novel object test was used to assess visual memory in rodents following established procedures. Animals were initially exposed visually to two identical objects, after which, one of the two objects was replaced with a new (novel) object. The time spent exploring each object was recorded. The test consisted of 3 phases: Habituation Phase: An animal was placed in an empty arena for 10 min to acclimate to the empty environment; Familiarization Phase: After spending 15 min back in the home cage, the animals re-entered the arena, now with two identical objects in place. The animal’s position in the arena was tracked for 3 min before returning to the home cage. Novel Object Phase: One of two objects was now replaced with a novel object, and the animal entered the arena and its position was again tracked for 3 min. Test took place twice, first at 1 h and then at 24 h after the introduction of the novel object to assess short-term and long-term memory, respectively [81].

### 4.11. Open Field Locomotion Test

The open field test utilized a large cubic box (measuring 1 m in length, width, and height) with an open top. An animal was placed at the center of the box, and its movements were video-recorded over several minutes to hours as it explored the environment. Computer tracking software (EthoVision XT software, Version 10.0; Noldus Information Technology, Wageningen, The Netherlands) analyzed their horizontal activity, time spent in different areas of the field, and total distance traveled.

### 4.12. Histological Examination

Following behavioral and electrophysiological assessments, 6 rats from each group were transcardially perfused with 4% paraformaldehyde in 0.1 M phosphate buffer (pH 7.4). Their paraspinous muscles were harvested and subjected to hematoxylin–eosin (H&E) staining, Sirius Red staining, Fast Green staining, and Masson’s trichrome staining. The muscle surface area was measured with the average of 20 randomly selected areas. The fibrosis ratio was calculated as the volume of red to green in the Sirius Red staining.

### 4.13. Statistical Analyses

Data were presented as mean ± standard error. Student’s *t*-test and repeated measures ANOVA were used to compare inter-group differences, whereas the post hoc analysis was conducted by Dunnett’s test. SPSS software version 12 was used for all statistical analyses, with a *p*-value < 0.05 considered statistically significant.

## 5. Conclusions

This study provided experimental evidence to bridge the knowledge gap in understanding the rationale behind IM steroid injections for preventing muscle injury and associated pain behaviors. We utilized a cell model combining hypoxia and LPS to simulate the pathological process of paraspinous muscle retraction, arising from ischemia and the subsequent inflammatory response. In the in vitro experiment, cell survival and the associated apoptosis cascade were restored. In the in vivo experiment, IM steroid injections effectively mitigated muscle injury and reduced inflammatory reactions in the dorsal root ganglion. These changes contributed to improvements in neurobehavioral outcomes and electrophysiological function. Based on the findings of this study, IM steroid injection appears to be a promising adjuvant approach in the management of post-spinal surgery care.

## Figures and Tables

**Figure 1 ijms-26-11093-f001:**
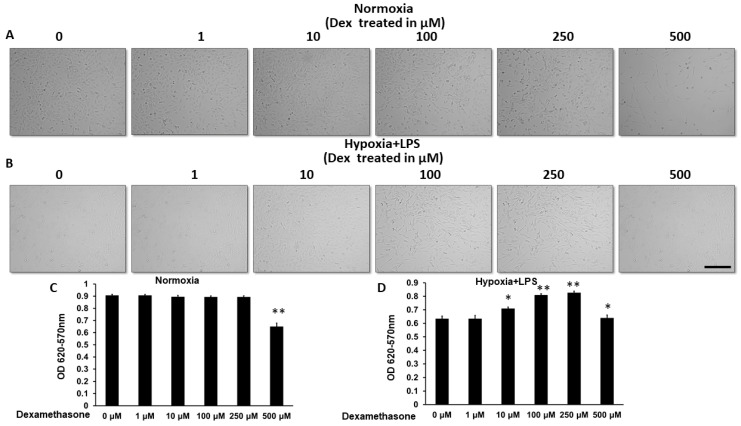
Cell viability measured by MTT under normoxia and hypoxia plus LPS-treated conditions with escalating doses of dexamethasone. The viability of C2C12 cells under normoxic and hypoxia + LPS conditions treated with escalating doses of dexamethasone was measured using MTT assays. (**A**) The cell morphology and density of C2C12 cells under normoxic conditions treated with escalating doses of dexamethasone, observed under a light microscope. (**B**) The cell morphology and density of C2C12 cells under hypoxia + LPS conditions treated with escalating doses of dexamethasone, observed under a light microscope. (**C**) Quantitative analysis of the optical density absorbance data from different treatment groups measured in (**A**). (**D**) Quantitative analysis of the optical density absorbance data from different treatment groups measured in (**B**). *N* = 3, based on three independent repeated measurements. Scale bar length = 1 mm. AR *: *p* < 0.05; **: *p* < 0.01.

**Figure 2 ijms-26-11093-f002:**
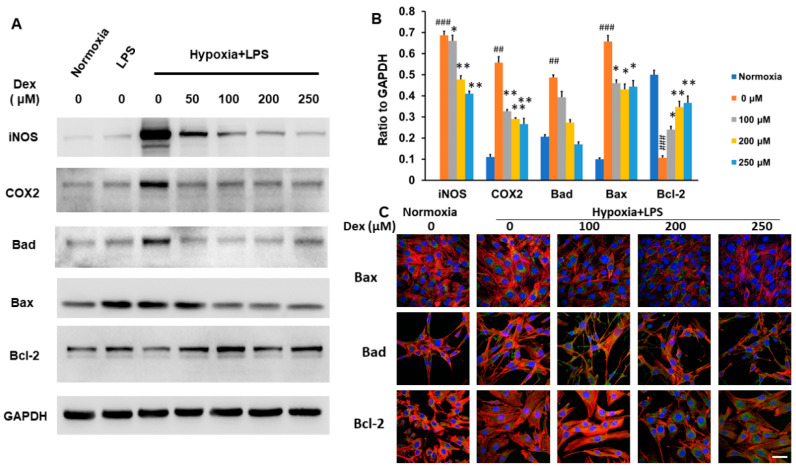
Modulation of Inflammatory Response and Apoptosis in C2C12 Cells under hypoxia + LPS Conditions by Dexamethasone. The effects of escalating doses of dexamethasone on C2C12 cells exposed to hypoxia + LPS conditions were analyzed using Western blot and immunohistochemistry staining. (**A**) Representative Western blot images showing protein expression levels under hypoxia + LPS conditions with varying doses of dexamethasone. (**B**) Quantitative analysis of the Western blot results from the different treatment groups shown in (**A**). (**C**) Immunohistochemistry staining for apoptosis markers (Bad, Bax, and Bcl-2) under hypoxia + LPS conditions with varying doses of dexamethasone. Data represent the mean of three independent experiments (*N* = 3). Red = Phalloidin. Blue = DAPI. Gree = Bax, Bad, Bcl-2. Scale bar length = 200 μm. Statistical significance: * *p* < 0.05; ** *p* < 0.01, compared to the 0 μM dexamethasone condition under hypoxia + LPS. ## *p* < 0.01; ### *p* < 0.001, compared to the normoxia condition.

**Figure 3 ijms-26-11093-f003:**
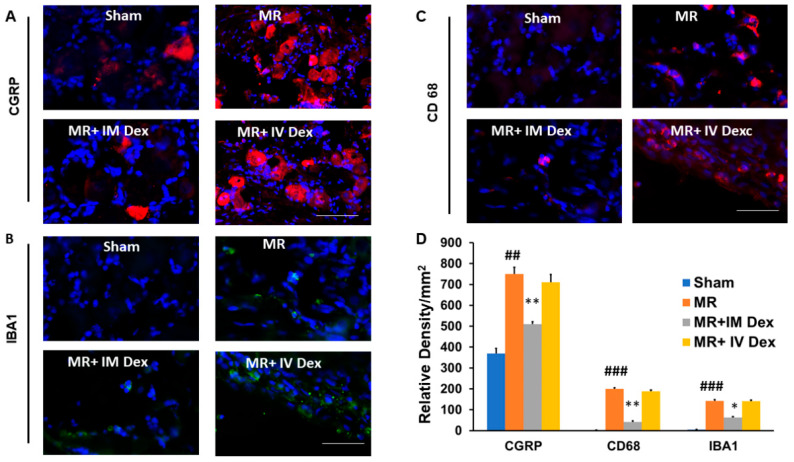
Immunohistochemistry Staining of CGRP and Inflammatory Cell Deposits in Dorsal Root Ganglion after Different Treatments. Animals underwent paraspinous muscle splitting with retractors to induce paraspinal muscle ischemia. They were allocated into four groups: sham, MR, MR+ IM dexamethasone, and MR+ IV dexamethasone. Dorsal root ganglion cells were harvested 7 days after the experiment and subjected to immunohistochemistry staining for CGRP, Iba-1, and CD68. (**A**) Representative CGRP staining in different treatment groups. Red = CGRP, Blue = DAPI. (**B**) Representative Iba-1 staining in different treatment groups. Green = Iba-1, Blue = DAPI. (**C**) Representative CD68 staining in different treatment groups. Red = CD68, Blue = DAPI. (**D**) Quantitative analysis of relative density of CGRP, Iba-1, and CD68 in different treatment groups. *N* = 3, bar length = 100 μm; *: *p* < 0.05; **: *p* < 0.001 indicate the *p*-value relative to the ischemia group. ##: *p* < 0.01; ###: *p* < 0.001 indicate the *p*-value relative to the sham group. Sham, MR, MR + IM Dex, MR + IV Dex: see text.

**Figure 4 ijms-26-11093-f004:**
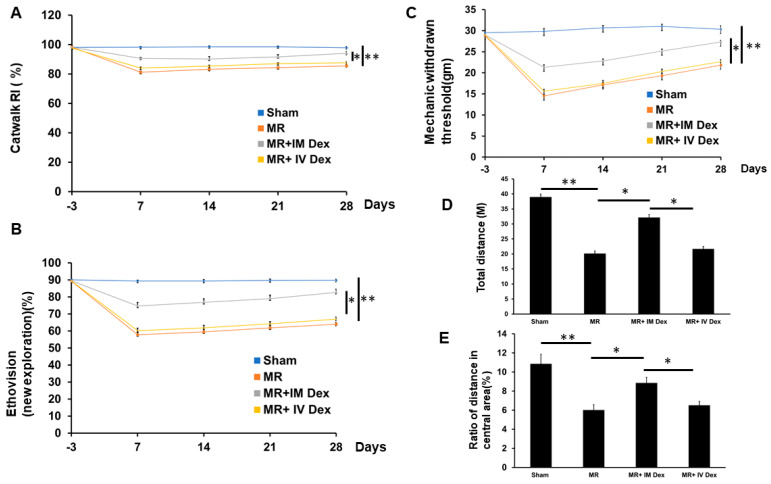
Neurobehavioral Assessment in Animals Subjected to Paraspinous Muscle Retraction Across Different Treatment Groups. Animals underwent paraspinous muscle splitting with retractors to induce paraspinal muscle ischemia and inflammation and were then allocated into four groups: sham, MR, MR+ IM dexamethasone, and MR + IV dexamethasone. Neurobehavioral assessments included Catwalk gait analysis, Ethovision, Von Frey test, and open field assessment. (**A**) The plot of the regularity index presented in percentage across different treatment groups. (**B**) Ratio of new exploration presented in Ethovision across different treatment groups. (**C**) The threshold of mechanical withdrawal across different treatment groups. (**D**) Bar graph showing total distance traveled across different treatment groups. (**E**) Bar graph showing distance traveled in the central area across different treatment groups. *N* = 6, *: *p* < 0.05; **: *p* < 0.01. Sham, MR, MR + IM Dex, MR + IV Dex: see text.

**Figure 5 ijms-26-11093-f005:**
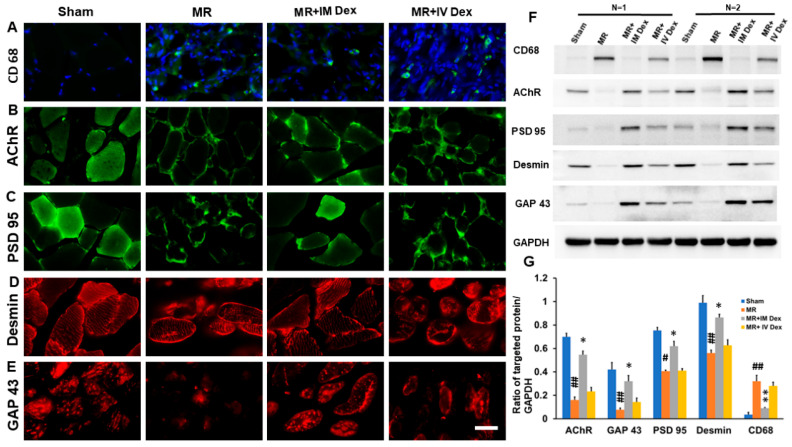
Increased Repair in Injured Muscle Following Intramuscular Injection of Dexamethasone. Animals underwent paraspinous muscle splitting with retractors to induce paraspinal muscle ischemia. They were divided into four groups: sham, MR, MR + IM dexamethasone, and MR + IV dexamethasone. Seven days after injury, the paraspinous muscles were harvested for immunohistochemistry staining and Western blot analysis. (**A**) Representative immunohistochemistry staining of CD 68 in the different treatment groups. (**B**) Representative immunohistochemistry staining of AChR in the different treatment groups. (**C**) Representative immunohistochemistry staining of PSD-95 in the different treatment groups. (**D**) Representative immunohistochemistry staining of Desmin in the different treatment groups. (**E**) Representative immunohistochemistry staining of GAP 43 in the different treatment groups. (**F**) Representative Western blot of CD 68, AChR, PSD-95, Desmin, and GAP-43 in the different treatment groups. (**G**) Quantitative analysis of Western blot with GAPDH as an internal control. #: *p* < 0.05; ##: *p* < 0.01 indicate *p*-values relative to the sham group. *: *p* < 0.05; **: *p* < 0.01 indicates *p*-values relative to the ischemia group. *N* = 3. Scale bar length = 200 μm. Sham, MR, MR + IM Dex, MR + IV Dex: see text.

**Figure 6 ijms-26-11093-f006:**
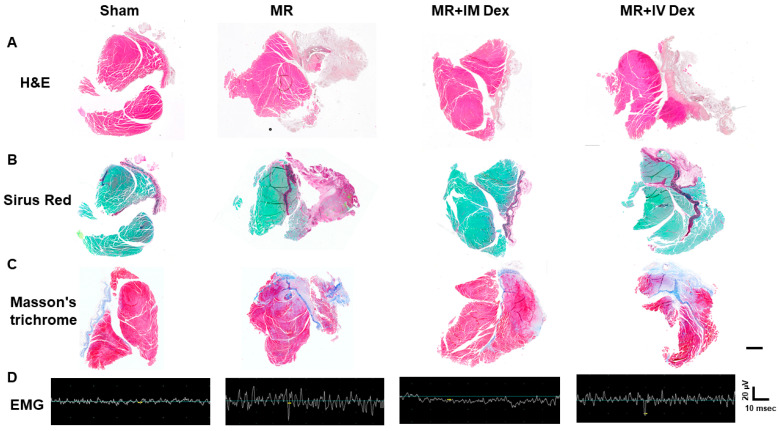
Attenuation of Muscle Fibrosis and Decreased Muscle Fibrillation After Intramuscular (IM) Dexamethasone. Animals underwent paraspinous muscle splitting with retractors to induce paraspinal muscle ischemia. They were divided into four groups: sham, MR, MR + IM dexamethasone, and MR + intravenous (IV) dexamethasone. Twenty-eight days post-injury, the paraspinous muscles were first examined via EMG and subsequently harvested for immunohistochemical staining. (**A**) H&E staining of the entire paraspinous muscle harvested from the animals. Scale bar = 20 μm. (**B**) Sirius Red staining of the paraspinous muscle at corresponding locations. Scale bar = 20 μm. (**C**) Masson’s trichrome staining of the paraspinous muscle at corresponding locations. Scale bar = 20 μm. (**D**) Representative EMG recordings from the different groups. Horizontal scale bar = 10 msec; vertical scale = 20 μV. Sham, MR, MR + IM Dex, MR + IV Dex: see text.

**Figure 7 ijms-26-11093-f007:**
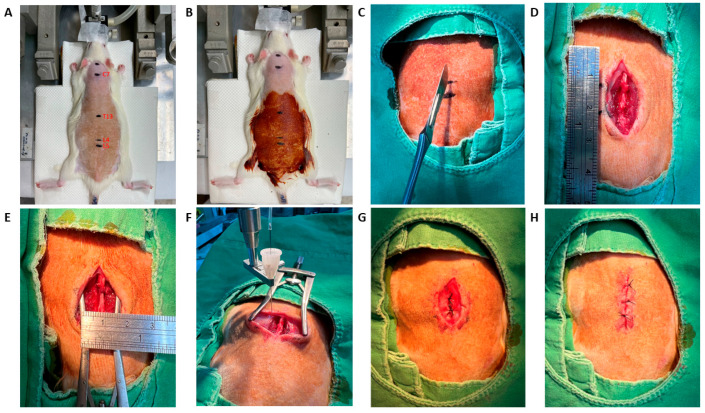
Procedure for Paraspinous Muscle Retraction. Animals underwent paraspinous muscle splitting with retractors to induce paraspinal muscle ischemia. (**A**) Spinal landmark for the operation. (**B**) Disinfect the wound. (**C**,**D**) Make a wound incision 3 cm in length at the midline. (**E**) Retract the wound to a width of 1.5 cm for 60 min. (**F**) Administer intramuscular injection of dexamethasone at 3 points on each side, with a total of 1 mg dexamethasone using an infusion pump. (**G**) Close the muscle in layers. (**H**) Close the surgical wound.

**Table 1 ijms-26-11093-t001:** Measurement of inflammatory cytokines in the paraspinous muscle in different treatment groups.

	Sham	MR	MR + IM Dex	MR + IV Dex	*p* Value
IL-β (pg/mL)	15.6 ± 2.3	500.3 ± 28.6	92.1 ± 7.6	320.3 ± 96.5	<0.0001
IL-6 (pg/mL)	15.3 ± 4.3	648.3 ± 29.1	203.3 ± 20.6	339.1 ± 35.2	<0.001
TNFα (pg/mL)	15.4 ± 4.1	3173.7 ± 196.4	136.3 ± 15.9	394.1 ± 8.9	<0.001

Data presented as mean ± standard errors. Sham, MR, MR + IM Dex, MR + IV Dex: see text.

**Table 2 ijms-26-11093-t002:** Measurement of size and fibrosis severity in the paraspinous muscle in different treatment groups.

	Sham	MR	MR + IM Dex	MR + IV Dex	*p* Value
Muscle surface area (μm^2^)	8088.3 ± 46.3	5436.7 ± 121.3	7346.7 ± 82.7	6103.1 ± 250.4	<0.0001
% of fibrosis to muscle	0.1 ± 0.06	36.1 ± 3.2	7.6 ± 0.3	14.7 ± 0.9	<0.001
Fiber density (Number/cm^2^)	26,188.7 ± 319.4	14,366.5 ± 426.7	23,038 ± 579.2	18,677.3 ± 416.6	<0.001

Data presented mean ± standard errors. Sham, MR, MR + IM Dex, MR + IV Dex: see text.

## Data Availability

The original contributions presented in this study are included in the article. Further inquiries can be directed to the corresponding author.

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
