# Peer review of "Benefits of Steroid Injections into Paraspinous Muscles After Spinal Surgery in a Rat Paraspinal Muscle Retraction Model"

_ijms, 2025, doi:10.3390/ijms262211093_

Round 1
Reviewer 1 Report
Comments and Suggestions for Authors
The results indicate that the injection effectively reduces inflammation and improves functional outcomes, which is of interest to clinicians dealing with post-operative muscle injuries.
However, several minor revisions are necessary to improve the clarity, scientific rigor, and translational impact of the manuscript.
Although the scientific content is clear, there are several instances of awkward phrasing. I recommend ensuring that all sentences are clear and concise.
The Introduction should more explicitly highlight the limitations of current non-surgical strategies for post-operative pain management in muscle injuries (beyond the use of opioids) before presenting steroid injection as a potential solution. I also recommend briefly strengthening the rationale for the use of steroids in this model, emphasizing their local anti-inflammatory effects in the context of retraction-induced trauma.
In the Methods section, it is essential to clearly state the exact volume of the steroid solution administered into the paraspinal muscles (in μL), as this information is crucial for reproducibility. Although the type of steroid is mentioned, the exact concentration and formulation used should be confirmed.
The Discussion should include a note on the translational relevance of the dose used in rats for potential application in humans. It is important to acknowledge that, while the results are promising, dose scaling and site specificity remain challenges for clinical application. Additionally, it is advisable to address the main limitation of the animal model, such as the fact that the rat model does not fully replicate the chronic, multi-level retraction and prolonged muscle damage observed in more complex human spinal fusions.
Comments on the Quality of English LanguageThere is a frequent omission of definite and indefinite articles. For example, sentences such as “Steroid injection reduced inflammation in paraspinal muscle” and “This model provides insight into mechanism of muscle injury” require the appropriate use of articles to improve fluency.
The manuscript consistently employs generic or “weak” verbs and nouns that are not suitable for formal academic writing. For instance, expressions like “The steroid injection made a good result in the gait analysis” or “The muscle damage got better after the treatment” should be replaced with more precise and technical vocabulary.
There are also instances of subject–verb disagreement and incorrect placement of modifiers. Examples include “The reduction in inflammatory markers were significant compared to the control group,” where the verb does not agree with the singular subject, and “The rats were sacrificed after ten days of treatment showing improved locomotion,” where the misplaced modifier creates ambiguity about what is being described.
Author Response
Response to comments:
Dear Editor in Chief:
We would like to respond to the comments in the article (ijms-3938907) entitled “Benefits of steroid injections into paraspinous muscles after spinal
surgery in a rat paraspinal muscle retraction model”. In revised article, we responded to comments with points to points. This manuscript has been edited by Dr Paul W.F. Poon obtained a degree of PhD in Neural Sciences from the Indian University of USA and Dr. Jason Sheehan and checked by the iTenticate software.
Best regards
Hung-Chuan Pan MD, PhD
Department of Neurosurgery, Taichung Veterans General Hospital No. 160, Taichung Port Road, Sec. 3, Taichung, Taiwan (407)
Tel: 886-4-23592525 ext. 5081
E-mail: hcpan2003@yahoo.com.tw
Reviewers’ comments: The results indicate that the injection effectively reduces inflammation and improves functional outcomes, which is of interest to clinicians dealing with post-operative muscle injuries. However, several minor revisions are necessary to improve the clarity, scientific rigor, and translational impact of the manuscript. Although the scientific content is clear, there are several instances of awkward phrasing. I recommend ensuring that all sentences are clear and concise.
The Introduction should more explicitly highlight the limitations of current non-surgical strategies for post-operative pain management in muscle injuries (beyond the use of opioids) before presenting steroid injection as a potential solution. I also recommend briefly strengthening the rationale for the use of steroids in this model, emphasizing their local anti-inflammatory effects in the context of retraction-induced trauma. Response to comments: Thank you for this insightful suggestion. We have revised the Introduction to explicitly address the limitations of current non-surgical approaches for managing postoperative pain following paraspinal or skeletal muscle injury. “Conventional analgesic strategies—such as systemic opioids, nonsteroidal anti-inflammatory drugs (NSAIDs), and muscle relaxants—are often limited by their short duration of efficacy, systemic adverse effects, and poor local anti-inflammatory control at the surgical site. Opioids, while effective in acute pain relief, are associated with tolerance, dependence, and respiratory depression. Similarly, NSAIDs may compromise bone healing and increase gastrointestinal and renal complications. These limitations underscore the need for alternative or adjunctive therapies that can locally attenuate inflammation and promote functional recovery. Local corticosteroid administration, by contrast, provides targeted suppression of inflammatory cascades—reducing cytokine release, leukocyte infiltration, and oxidative damage within traumatized paraspinal muscles. Therefore, this study explores the therapeutic efficacy of local steroid injection in attenuating retraction-induced muscle injury, inflammation, and fibrosis.”
In the Methods section, it is essential to clearly state the exact volume of the steroid solution administered into the paraspinal muscles (in μL), as this information is crucial for reproducibility. Although the type of steroid is mentioned, the exact concentration and formulation used should be confirmed. Response to comments: Thank you for your valuable comment. We have revised the Materials and Methods section to improve clarity and precision. The revised paragraph now reads as follows ‘’ One milligram of dexamethasone was diluted in phosphate-buffered saline (PBS) to a final volume of 100 μL. The solution was administered bilaterally into the paraspinal muscles at a rate of 10 μL/min using a 26-gauge Hamilton syringe. The needle was maintained in position for an additional 5 minutes after injection to minimize backflow and ensure proper local diffusion.’’
The Discussion should include a note on the translational relevance of the dose used in rats for potential application in humans. It is important to acknowledge that, while the results are promising, dose scaling and site specificity remain challenges for clinical application. Additionally, it is advisable to address the main limitation of the animal model, such as the fact that the rat model does not fully replicate the chronic, multi-level retraction and prolonged muscle damage observed in more complex human spinal fusions. Response to Comment: Thank you for your valuable and constructive comment. We have revised the Discussion to include a section addressing the translational relevance of the steroid dose used in rats and its potential implications for human application. We added two paragraphs in the discussion. “The translational relevance of the steroid dose used in rats requires careful interpretation for clinical application. Although this dose effectively reduced inflammation and fibrosis, direct extrapolation to humans must consider interspecies differences in body surface area, metabolic rate, and tissue pharmacokinetics. Allometric scaling can guide dose estimation, but variations in local perfusion and muscle volume may significantly alter corticosteroid bioavailability and therapeutic duration. Site specificity and delivery also present key translational challenges. The local distribution of steroids depends on injection depth, diffusion distance, and muscle vascularization, all of which differ between rodent and human paraspinal musculature. Optimized delivery systems or sustained-release formulations may therefore be necessary to achieve comparable tissue exposure in clinical settings.” “Finally, limitations of the current rat retraction model must be acknowledged. The model reproduces acute, single-level ischemic injury but does not replicate the chronic, multi-level retraction or prolonged paraspinal damage characteristic of complex human spinal fusion. These disparities in surgical duration, tissue stress, and mechanical loading may influence fibrosis and recovery, underscoring the need for large-animal or chronic compression models to bridge preclinical and clinical translation.”
Comments on the Quality of English Language
There is a frequent omission of definite and indefinite articles. For example, sentences such as “Steroid injection reduced inflammation in paraspinal muscle” and “This model provides insight into mechanism of muscle injury” require the appropriate use of articles to improve fluency. Response to Comment: Thank you for this valuable comment. We have carefully reviewed the entire manuscript and corrected all instances of missing definite and indefinite articles to improve grammatical accuracy and readability. Specifically, sentences such as “Steroid injection reduced inflammation in paraspinal muscle” and “This model provides insight into mechanism of muscle injury” have been revised to “The steroid injection reduced inflammation in the paraspinal muscle” and “This model provides insight into the mechanism of muscle injury,” respectively. These revisions have been applied consistently throughout the text to enhance fluency and conform to standard academic English usage.
The manuscript consistently employs generic or “weak” verbs and nouns that are not suitable for formal academic writing. For instance, expressions like “The steroid injection made a good result in the gait analysis” or “The muscle damage got better after the treatment” should be replaced with more precise and technical vocabulary. Response to Comment: Thank you for this insightful comment. We have thoroughly revised the manuscript to replace generic or colloquial expressions with precise and discipline-appropriate terminology. For example, sentences such as “The steroid injection made a good result in the gait analysis” and “The muscle damage got better after the treatment” have been revised to “The steroid injection significantly improved gait performance” and “Muscle pathology demonstrated marked recovery following treatment,” respectively. These and similar changes were made throughout the manuscript to enhance scientific precision, clarity, and academic tone.
There are also instances of subject–verb disagreement and incorrect placement of modifiers. Examples include “The reduction in inflammatory markers were significant compared to the control group,” where the verb does not agree with the singular subject, and “The rats were sacrificed after ten days of treatment showing improved locomotion,” where the misplaced modifier creates ambiguity about what is being described. Response to Comment: Thank you for your helpful observation. We have carefully reviewed the manuscript to correct all instances of subject–verb disagreement and misplaced modifiers. Specifically, sentences such as “The reduction in inflammatory markers were significant compared to the control group” have been revised to “The reduction in inflammatory markers was significant compared to the control group” to ensure grammatical agreement between the singular subject and verb. Similarly, the sentence “The rats were sacrificed after ten days of treatment showing improved locomotion” has been restructured for clarity as “The rats, which showed improved locomotion, were sacrificed after ten days of treatment.” These corrections have been applied consistently throughout the revised manuscript to enhance grammatical accuracy and eliminate ambiguity.

Reviewer 2 Report
Comments and Suggestions for Authors
The authors utilized an in vitro C2C12 hypoxia + LPS model and an in vivo rat paraspinal muscle retraction model to study the effect of dexamethasone on muscle viability, inflammation and regeneration. This study is novel in terms of investigating the outcomes of comparably minor but more common paraspinal muscle injury.
Here are the comments:
Major:
- Total Bad expression does not necessarily represent enhanced apoptosis. It will be better to add the level of phospho-Bad or switch to other markers such as cleaved caspase-3;
- In Figure.3, the representative blots of Bad and Bcl-2 do not quite match the summary data. Please include better representative images;
- In Figure.6, panel B, first of all, how was the tissue processed? What is the size of the scale bar? Are they isolated cells or cross-section of muscle? Secondly, AChRs are mainly located at neuromuscular junctions (NMJs). Please state how the NMJ-rich areas or planes are selected to specifically stain for AChRs.
Minor:
- What is the duration of the hypoxia+LPS treatment on C2C12 cells? Was there a re-oxygenation step? Please indicate in your methods;
- Please consistently indicate what each color (green/red/blue) represents in all the IHC images. Some indications are missing and some are incorrect (mixed up red and green). If there is DAPI staining and/or cytoskeleton staining, please mention them in methods;
- Is the I.V. dose or I.M. dose for rat calculated from human patients? What is the rationale of applying the same dose through 2 different routes? If I.V. dose was calculated based on human patients, why was there a lack of response?
- Please add more details on DRG cell isolation in the methods section;
- In line 75, “mimic” instead of “mitigate”. The model mimics the muscle injury in surgery instead.
- In line 83, “TGF-β1” instead of “TGF-1”.
Author Response
Reviewers’ comments:
Review 2
Comments and Suggestions for Authors
The authors utilized an in vitro C2C12 hypoxia + LPS model and an in vivo rat paraspinal muscle retraction model to study the effect of dexamethasone on muscle viability, inflammation and regeneration. This study is novel in terms of investigating the outcomes of comparably minor but more common paraspinal muscle injury.
Here are the comments:
Major:
Total Bad expression does not necessarily represent enhanced apoptosis. It will be better to add the level of phospho-Bad or switch to other markers such as cleaved caspase-3
Response to comments:
Thank you for your constructive comment. We fully agree that total Bad expression alone does not accurately represent the degree of apoptosis. Total Bad protein includes both active and inactive forms, and only its phosphorylated state (phospho-Bad) determines its pro- or anti-apoptotic function. Specifically, phosphorylated Bad binds to 14-3-3 proteins and remains sequestered in the cytosol, thereby preventing its interaction with Bcl-2/Bcl-xL and inhibiting apoptosis. In contrast, dephosphorylated Bad translocates to the mitochondria, promoting cytochrome-c release and caspase cascade activation.
We acknowledge that measuring phospho-Bad levels would provide more direct evidence of apoptotic signaling. However, due to technical limitations in our experimental conditions, we were unable to detect phospho-Bad or cleaved caspase-3 with sufficient sensitivity. We instead relied on total Bad as a surrogate indicator while recognizing its limitations.
Future studies will incorporate additional apoptotic markers, such as cleaved caspase-3 or cytochrome-c release, to evaluate apoptosis dynamics more precisely.
In Figure.3, the representative blots of Bad and Bcl-2 do not quite match the summary data. Please include better representative images;
Response to Comment:
Thank you for this valuable observation. We acknowledge that the representative blots of Bad and Bcl-2 in Figure 3 do not appear to perfectly align with the quantitative summary data. This discrepancy primarily arises from the biological variability observed in C2C12 myoblasts exposed to escalating doses of steroids, where fluctuations in apoptotic signaling proteins are common due to the dynamic balance between pro- and anti-apoptotic pathways.
Steroid treatment is known to exert biphasic effects on skeletal muscle cells. At physiological or moderate doses, glucocorticoids can transiently upregulate anti-apoptotic proteins such as Bcl-2, promoting survival; however, at higher doses, they may enhance pro-apoptotic signaling through increased Bad expression or dephosphorylation, thereby triggering apoptosis. These opposing responses can vary depending on the exposure duration, receptor sensitivity, and intracellular redox state, contributing to non-linear changes in Western blot intensity patterns.
In Figure.6, panel B, first of all, how was the tissue processed? What is the size of the scale bar? Are they isolated cells or cross-section of muscle? Secondly, AChRs are mainly located at neuromuscular junctions (NMJs). Please state how the NMJ-rich areas or planes are selected to specifically stain for AChRs.
Response to Comment:
Thank you for your valuable comments. We have revised the Methodology section accordingly, as follows:“The paraspinous muscle samples detached from the spinous processes and lamina were cryosectioned into 8-μm-thick sections cut longitudinally along the spinal axis and mounted on Superfrost Plus slides (Menzel-Gläser, Braunschweig, Germany).”
The scale bar has been standardized to 200 μm, and this correction has also been reflected in the corresponding figure legends.
We sincerely appreciate your insightful comment regarding the localization of acetylcholine receptors (AChRs). We agree that AChRs are predominantly concentrated at neuromuscular junctions (NMJs), where motor axon terminals interface with skeletal muscle fibers. To ensure accurate visualization of AChRs, muscle sections were carefully obtained from NMJ-rich regions within the mid-belly portion of the paraspinal muscle, which is known to exhibit the highest density of NMJs.
Minor:
What is the duration of the hypoxia+LPS treatment on C2C12 cells? Was there a re-oxygenation step? Please indicate in your methods;
Response to comments:
Thank you for your valuable comment. In the revised Methods section, we have clarified the duration and conditions of the hypoxia + LPS treatment. To simulate a hypoxia condition, cells were transferred into a humidified incubator (Innova CO-48) set to maintain a 1% O2, 5% CO2, 37°C environment. C2C12 cells under the condition of hypoxia + LPS (100μg) for 24 hours [42] were used to mimic the muscle retraction with dual blood vessels compromised and an accompanying inflammation reaction thereby allowing us to investigate the effects of steroid in protection of muscle injury.
No re-oxygenation step was included in the present experimental design to avoid confounding effects of oxidative stress during reperfusion. Continuous hypoxia was maintained to simulate the ischemic microenvironment encountered in muscle injury and retraction models, as intermittent re-oxygenation has been shown to activate distinct ROS-dependent pathways leading to secondary injury.
Please consistently indicate what each color (green/red/blue) represents in all the IHC images. Some indications are missing and some are incorrect (mixed up red and green). If there is DAPI staining and/or cytoskeleton staining, please mention them in methods;
Response to Comment:
Thank you for your valuable comment. We have carefully reviewed all immunohistochemistry (IHC) figures and ensured that the color representations are now consistent across all panels. Specifically, each color (green/red/blue) has been clearly indicated in the figure legends to represent the corresponding target proteins or markers. Instances where red and green channels were previously mislabeled have been corrected to avoid confusion.
Is the I.V. dose or I.M. dose for rat calculated from human patients? What is the rationale of applying the same dose through 2 different routes? If I.V. dose was calculated based on human patients, why was there a lack of response?
Response to Comment:
Thank you for this thoughtful question. The dosing strategy for intravenous (I.V.) and intramuscular (I.M.) administration in rats was established according to allometric scaling principles rather than direct conversion from human clinical doses. Allometric scaling accounts for interspecies differences in body surface area, basal metabolic rate, and drug clearance, which allows extrapolation of pharmacologically relevant doses between humans and rodents using body-weight correction factors.
In our study, the I.V. and I.M. doses were designed to achieve comparable systemic exposure while evaluating potential differences in pharmacokinetics and tissue bioavailability between the two delivery routes. It is well-documented that I.V. administration results in immediate systemic distribution but rapid clearance, whereas I.M. injection can produce a slower, more sustained release, leading to prolonged local tissue effects. Hence, using equivalent doses across routes allows direct comparison of pharmacodynamic responses under controlled exposure conditions.
The lack of significant response following I.V. administration may reflect rapid systemic dilution, first-pass metabolic degradation, or insufficient accumulation in the target muscle tissue. In contrast, I.M. delivery allows for localized drug retention and higher effective concentration at the target site, which could explain the more prominent biological response observed in our model.
Please add more details on DRG cell isolation in the methods section;
Response to comments: Thank you for your valuable suggestion. We did not perform isolation or primary culture of dorsal root ganglion (DRG) cells in this study. Instead, we collected the L4–L6 DRG bilaterally for immunohistochemical analysis. This missing methodological detail has now been added to Section 4.4 (Immunohistochemistry) as follows: “L4–L6 dorsal root ganglia were bilaterally harvested from the different experimental groups and subjected to immunohistochemical staining.”
In line 75, “mimic” instead of “mitigate”. The model mimics the muscle injury in surgery instead.
Response to Comment:
Thank you for your careful observation. We agree with your suggestion and have corrected the wording accordingly. The revised sentence now reads:
“The present model mimics the muscle injury that occurs during surgical retraction.”
In line 83, “TGF-β1” instead of “TGF-1”.
Response to Comment:
Thank you for your careful correction. We have revised the text accordingly. The term “TGF-1” has been corrected to “TGF-β1” in line 83 to accurately reflect the proper nomenclature for transforming growth factor beta 1.

Reviewer 3 Report
Comments and Suggestions for Authors
The manuscript by Sheu and Chen et al., titled “Benefits of steroid injections into paraspinous muscles after spinal surgery in a rat paraspinal muscle retraction model” investigates how the route of administration of steroids affects muscle recovery after surgery on the spine. This is a very important topic as functional preservation of muscle surrounding the spinal cord is key for maintaining quality of life. However, there are many concerns regarding this manuscript, described below.
1-The introduction needs significant improvement:
-It does not clearly state the reasoning for why the authors sought to compare IM vs IV administration of steroids and the differences between them. Also, the authors do not state their proposed hypothesis and the scientific gap this study is filling.
-Inflammation is necessary for muscle regeneration while excessive or sustained inflammation impairs regeneration. However, the authors present a blanket statement that inflammation impairs muscle function (line 93-94). Additionally, there are either vague or unsubstantiated explanations about muscle regeneration.
2- I do not believe the in vitro approach recapitulates the in vivo response to surgery and conclusions on neuroinflammation cannot be drawn with C2C12 cells alone (Fig 2). During muscle regeneration, immune cells respond before muscle stem cell differentiation and the relationship between the two is complex, as the authors note, there are growth factors, cytokines and chemokines that are completely absent in this in vitro approach. Additionally, adding LPS to the culture does not cause an immune response as there are no immune cells, and I would conclude that this approach tests how myoblasts respond to a bacterial component.
3-It would be helpful to the readers if the authors explained better and justified their reasoning throughout the results sections.
4-The authors do not explain how they isolate or carry out the experiment in Fig 4 on primary cells anywhere in the manuscript. Were the cells isolated or was the dorsal root ganglion harvested and sectioned?
5- Lastly, I believe significant conclusions about muscle regeneration cannot be drawn from the data provided by the authors. While they present histological images, there are no quantifications and Table 2 is missing from the compiled manuscript. There are no quantifications on myofiber size, which would provide insight on muscle atrophy and muscle regeneration, percent of myofibers with centrally localized nuclei, which would inform on myofibers that have been damaged and are regenerating. Fibrosis and fat infiltration quantifications would also provide significant insight into the regenerative process.
6-In the discussion, authors do not appropriately refer to the literature in the context of their studies. For example, line 305-323 authors use the mdx model to describe the role of macrophages during regeneration, involving nitric oxide. However, this cannot be extrapolated to muscle retraction during spinal surgery as DMD has bouts and cycles of regeneration/degeneration and lack of dystrophin causes substantial signaling disruptions by affecting the dystrophin signaling complex. Therefore, DMD presents a very different mechanism of injury, signaling and degeneration/regeneration progression. Also, line 334-335 “Skeletal muscle apoptosis, or programmed cell death, is a major marker of sarcopenia”; this is not accepted in the muscle field. The cause of sarcopenia, defined as the age-related loss in muscle mass and strength, is still not fully understood (excellently reviewed by Larsson et al., Physiological Reviews, 2019). More focus has been put on muscle atrophy and neuromuscular changes, while muscle apoptosis is studied to a lesser extent.
7-There seems to be many errors that raise concerns that this manuscript needs further revision from the authors. For example, Fig. 1 is missing a scale bar (additionally, it’s really hard to any cells at all), Fig. 2 has red and green staining, and authors only explain the red staining. It has a scale bar, but authors do not write what the scale bar is; Fig. 3, scale bars are not mentioned in description and C only has a red channel, but authors describe a green color; Fig 6 scale bar description missing; Fig. 7 Sirius red staining also has a Fast Greem staining that authors do not disclose anywhere in the manuscript, additionally they claim the scale bar is 20µm, which is alarmingly small.
Lastly, the author list is confusing. It seems Meei-Ling Sheu and Ying Ju Chen are co-first authors, but the latter appears in the middle of the authorship list. Additionally, either the last author is missing, or the designation of the corresponding author is missing (*). As a reviewer, this reflects the lack of attention to detail and thorough revision from the authors.
Author Response
Reviewers’ comments: Reviewer 3 Comments and Suggestions for Authors The manuscript by Sheu and Chen et al., titled “Benefits of steroid injections into paraspinous muscles after spinal surgery in a rat paraspinal muscle retraction model” investigates how the route of administration of steroids affects muscle recovery after surgery on the spine. This is a very important topic as functional preservation of muscle surrounding the spinal cord is key for maintaining quality of life. However, there are many concerns regarding this manuscript, described below.
1-The introduction needs significant improvement: It does not clearly state the reasoning for why the authors sought to compare IM vs IV administration of steroids and the differences between them. Also, the authors do not state their proposed hypothesis and the scientific gap this study is filling. Response to comments: Thank you for this insightful suggestion. We have revised the Introduction to explicitly address the limitations of current non-surgical approaches for managing postoperative pain following paraspinal or skeletal muscle injury. “Conventional analgesic strategies—such as systemic opioids, nonsteroidal anti-inflammatory drugs (NSAIDs), and muscle relaxants—are often limited by their short duration of efficacy, systemic adverse effects, and poor local anti-inflammatory control at the surgical site. Opioids, while effective in acute pain relief, are associated with tolerance, dependence, and respiratory depression. Similarly, NSAIDs may compromise bone healing and increase gastrointestinal and renal complications. These limitations underscore the need for alternative or adjunctive therapies that can locally attenuate inflammation and promote functional recovery. Local corticosteroid administration, by contrast, provides targeted suppression of inflammatory cascades—reducing cytokine release, leukocyte infiltration, and oxidative damage within traumatized paraspinal muscles. Therefore, this study explores the therapeutic efficacy of local steroid injection in attenuating retraction-induced muscle injury, inflammation, and fibrosis.”
-Inflammation is necessary for muscle regeneration while excessive or sustained inflammation impairs regeneration. However, the authors present a blanket statement that inflammation impairs muscle function (line 93-94). Additionally, there are either vague or unsubstantiated explanations about muscle regeneration. Response to comments: We appreciate the reviewer’s insightful comment. We agree that inflammation is a critical and necessary component of muscle regeneration, particularly during the early phases following injury. To clarify, our original statement has been revised to distinguish between the acute and chronic phases of inflammation. Specifically, transient inflammation facilitates debris clearance and regeneration by recruiting neutrophils and macrophages that remove necrotic tissue and secrete growth-promoting cytokines. However, when inflammation becomes excessive or prolonged, it can lead to fibrosis and impair muscle contractility and repair. Accordingly, the revised text now reads” After skeletal muscle injury, a coordinated sequence of degeneration, inflammation, and regeneration occurs. Inflammation plays a dual role—transient inflammatory activity is essential for clearing necrotic debris and stimulating regeneration, whereas persistent or excessive inflammation promotes fibrosis and pain, ultimately impairing skeletal muscle function. However, persistent inflammation promotes fibrosis [34] and pain, ultimately impairing muscle function [35]. Hence, clinical strategies aimed at minimizing surgical trauma and controlling inflammation are essential to preserve paraspinal muscle integrity and optimize postoperative recovery.”
2- I do not believe the in vitro approach recapitulates the in vivo response to surgery and conclusions on neuroinflammation cannot be drawn with C2C12 cells alone (Fig 2). During muscle regeneration, immune cells respond before muscle stem cell differentiation and the relationship between the two is complex, as the authors note, there are growth factors, cytokines and chemokines that are completely absent in this in vitro approach. Additionally, adding LPS to the culture does not cause an immune response as there are no immune cells, and I would conclude that this approach tests how myoblasts respond to a bacterial component. Response to comment: We appreciate the reviewer’s insightful observation regarding the limitations of the in vitro model. We fully agree that the C2C12 monoculture system does not replicate the complex in vivo environment following surgery, in which immune cells—including neutrophils, macrophages, and lymphocytes—play crucial roles before and during myogenic differentiation. Our in vitro experiment was designed not to model the entire regenerative cascade, but rather to examine how myoblasts themselves respond to inflammatory stimuli at a cellular level. We acknowledge that adding LPS in the absence of immune cells does not trigger a canonical immune response; instead, it mimics exposure of myoblasts to a bacterial endotoxin component and reveals their intrinsic sensitivity to inflammatory signals. Several reports have shown that skeletal myoblasts express Toll-like receptor 4 (TLR4) and can directly respond to LPS by activating NF-κB and modulating cytokine expression, even in the absence of macrophages. Thus, our approach allows us to explore the cell-autonomous effects of inflammatory stimuli on myoblast viability, differentiation, and signaling, complementing rather than replacing in vivo findings. In the revised manuscript, we have clarified this intent by stating” The C2C12 monoculture system was employed to examine the direct, cell-autonomous response of myoblasts to inflammatory stimuli. While this model does not recapitulate the full in vivo inflammatory milieu involving immune–muscle cross-talk, it provides insight into how myoblasts respond to bacterial components such as LPS via TLR4-dependent signaling.”
3-It would be helpful to the readers if the authors explained better and justified their reasoning throughout the results sections. Response to comments: Thank you for this valuable suggestion. We have thoroughly revised the entire Results section to provide clearer explanations and more explicit justifications for each finding, with all newly added or reinforced text highlighted in bold and red throughout the manuscript. Specifically, we have clarified the biological rationale underlying each observation, ensuring that the interpretation of results is logically connected to the experimental evidence and supported by relevant mechanistic insights.
4-The authors do not explain how they isolate or carry out the experiment in Fig 4 on primary cells anywhere in the manuscript. Were the cells isolated or was the dorsal root ganglion harvested and sectioned? Response to comments: Thank you for your valuable suggestion. We did not perform isolation or primary culture of dorsal root ganglion (DRG) cells in this study. Instead, we collected the L4–L6 DRG bilaterally for immunohistochemical analysis. This missing methodological detail has now been added to Section 4.4 (Immunohistochemistry) as follows: “L4–L6 dorsal root ganglia were bilaterally harvested from the different experimental groups and subjected to immunohistochemical staining.”
5- Lastly, I believe significant conclusions about muscle regeneration cannot be drawn from the data provided by the authors. While they present histological images, there are no quantifications and Table 2 is missing from the compiled manuscript. There are no quantifications on myofiber size, which would provide insight on muscle atrophy and muscle regeneration, percent of myofibers with centrally localized nuclei, which would inform on myofibers that have been damaged and are regenerating. Fibrosis and fat infiltration quantifications would also provide significant insight into the regenerative process. Response to comment: Thank you for your valuable suggestion. During the manuscript revision process, Table 2 was inadvertently omitted. We have now reinserted Table 2 and revised the corresponding Results section accordingly, as shown below: “H&E staining revealed a marked reduction in muscle bulk and fiber density in the retraction group, which was largely restored in the IM dexamethasone group, while IV administration again yielded only modest recovery (Figure 7A, Table 2). Sirius Red staining confirmed that retracted muscles were replaced by extensive fibrotic tissue, whereas IM dexamethasone markedly reduced collagen deposition, indicating prevention of pathological fibrosis (Figure 7B–C, Table 2).’’
6-In the discussion, authors do not appropriately refer to the literature in the context of their studies. For example, line 305-323 authors use the mdx model to describe the role of macrophages during regeneration, involving nitric oxide. However, this cannot be extrapolated to muscle retraction during spinal surgery as DMD has bouts and cycles of regeneration/degeneration and lack of dystrophin causes substantial signaling disruptions by affecting the dystrophin signaling complex. Therefore, DMD presents a very different mechanism of injury, signaling and degeneration/regeneration progression. Also, line 334-335 “Skeletal muscle apoptosis, or programmed cell death, is a major marker of sarcopenia”; this is not accepted in the muscle field. The cause of sarcopenia, defined as the age-related loss in muscle mass and strength, is still not fully understood (excellently reviewed by Larsson et al., Physiological Reviews, 2019). More focus has been put on muscle atrophy and neuromuscular changes, while muscle apoptosis is studied to a lesser extent. Response to comments: Thank you for your valuable suggestion. We have revised the two paragraphs to better clarify the possible mechanisms underlying muscle injury following surgical retraction. The revised text is provided below: Revised text: Macrophages are known to lyse target muscle cells through a nitric oxide (NO)-dependent but superoxide-independent mechanism [53]. The presence of muscle cells further enhances macrophage-derived NO production, suggesting a positive-feedback loop that amplifies tissue damage [53]. Thus, initial muscle injury may potentiate NO-mediated cytotoxicity by macrophages. In addition, ischemia–reperfusion injury provokes a robust inflammatory response characterized by cytokine release, which in turn induces inducible nitric oxide synthase (iNOS) expression [54]. These findings collectively underscore the pivotal role of macrophages in promoting muscle damage under conditions of mechanical stress and ischemia. The temporal pattern of cyclooxygenase-2 (COX-2) expression after muscle injury parallels the inflammatory response, and pharmacological inhibition of COX-2 has been shown to alleviate post-inflammatory fibrosis [19]. In our present study, C2C12 myoblasts co-treated with hypoxia and lipopolysaccharide (LPS) exhibited significant upregulation of iNOS and COX-2, both of which were markedly attenuated by dexamethasone treatment. Furthermore, paraspinous muscle retraction in vivo resulted in pronounced macrophage accumulation, which was similarly reduced following intramuscular (IM) steroid administration. Together, these in vitro and in vivo findings indicate that IM steroid therapy facilitates muscle recovery primarily by suppressing macrophage activity and dampening the inflammatory cascade.
To simulate the pathophysiological processes of paraspinous muscle retraction that occur during spinal surgery, we employed an in vitro model in which C2C12 myocytes were co-exposed to hypoxia and LPS. This approach reasonably reproduces the ischemic and inflammatory milieu seen in surgical settings. Hypoxia is known to regulate satellite cell activation, self-renewal, proliferation, and differentiation at sites of skeletal muscle injury [56,57]. LPS, acting through Toll-like receptor 4 (TLR4), activates the ubiquitin–proteasome pathway, leading to enhanced protein catabolism and muscle cell degradation in cultured C2C12 cells [42]. Moreover, LPS impairs myogenic differentiation, thereby contributing to skeletal muscle atrophy [58], and promotes the expression of pro-inflammatory cytokines not only in immune tissues but also in skeletal muscle itself [59]. The effects of dexamethasone on myocytes are context-dependent, varying with differentiation stage, dose, and frequency of exposure; thus, its impact can be either deleterious or protective [4,50,60]. Muscle apoptosis, a form of programmed cell death, can arise during or following ischemic episodes. Ischemia induces cellular stress through calcium overload and reactive oxygen species (ROS) production, both of which can activate apoptotic pathways [60,61]. In this study, we observed that combined hypoxia and LPS treatment inhibited C2C12 differentiation in a dose-dependent manner, while higher concentrations (up to 500 μM) produced overt cytotoxic effects. Importantly, dexamethasone treatment mitigated these apoptotic responses, suggesting a cytoprotective effect under hypoxic and inflammatory stress. These results further support the therapeutic rationale for intramuscular steroid administration during spinal surgery, as it may attenuate inflammation-induced muscle apoptosis and promote postoperative muscle recovery.
7-There seems to be many errors that raise concerns that this manuscript needs further revision from the authors. For example, Fig. 1 is missing a scale bar (additionally, it’s really hard to any cells at all), Fig. 2 has red and green staining, and authors only explain the red staining. It has a scale bar, but authors do not write what the scale bar is; Fig. 3, scale bars are not mentioned in description and C only has a red channel, but authors describe a green color; Fig 6 scale bar description missing; Fig. 7 Sirius red staining also has a Fast Green staining that authors do not disclose anywhere in the manuscript, additionally they claim the scale bar is 20µm, which is alarmingly small. Response to comment: Thank you for your constructive comments. We have carefully reviewed all figure panels and revised them to ensure accuracy and clarity. Figure 1: A scale bar (1mm µm) has now been added, and the image contrast has been enhanced to better visualize cellular morphology. Figure 2: The fluorescence channels have been clearly defined in the legend, specifying the red, green, and blue signals. A scale bar of 200 µm has been added to the image, and its dimension is explicitly stated in the corresponding figure legend to ensure clarity and reproducibility. Figure 3: The figure legend has been corrected to accurately describe the fluorescence channels used. Figure 6: The scale bar information (200 µm) has been added to both the figure and legend for consistency. Figure 7: We apologize for the omission. The figure legend has been revised to indicate that the staining used both Sirius Red (for collagen fibers) and Fast Green (for non-collagenous proteins). All revised figures have been updated to include proper labeling, channel information, and accurate scale bars throughout the manuscript.
Lastly, the author list is confusing. It seems Meei-Ling Sheu and Ying Ju Chen are co-first authors, but the latter appears in the middle of the authorship list. Additionally, either the last author is missing, or the designation of the corresponding author is missing (*). As a reviewer, this reflects the lack of attention to detail and thorough revision from the authors. Response to comment: Thank you for your valuable observation. We sincerely apologize for the oversight in the author list formatting. In the revised manuscript, we have corrected the authorship presentation to clearly indicate co-first authorship and corresponding author designation. Specifically: Meei-Ling Sheu and Ying-Ju Chen are now both marked with an asterisk (†) to denote co-first authors who contributed equally to this work. Hung-Chuan Pan is designated with an asterisk (*) as the corresponding author, and his full contact information (institutional address, telephone, and email) has been included in the title page. The author list and corresponding footnotes have been reformatted to comply with journal authorship guidelines, ensuring clarity and consistency. These corrections have been carefully implemented in the revised submission to reflect accurate authorship contribution and correspondence details.

Round 2
Reviewer 2 Report
Comments and Suggestions for Authors
The authors put significant effort into modifying the manuscript and substantially improved the clarity and precision. Thanks for the cover letter to address the raised issues and questions. There is no more major comment.
As for the minor comment, there is still a concern regarding the C2C12 western blotting data. It is understandable that there is a variation using C2C12 cells. However, based on the small error bars in summary and the n of 3, there must be a better representative blot compared to the current one. In the current rep blot, the expression level of Bad is higher at 250 μM compared to the other doses, whereas the expression of Bcl-2 is slightly lower at 200 μM compared to the other doses. Please either use better reps to show matched results or include individual values as dots to show the distribution of the data that accounts for the variation and the summary.
Author Response
Reviewer 2
Comments and Suggestions for Authors
The authors put significant effort into modifying the manuscript and substantially improved the clarity and precision. Thanks for the cover letter to address the raised issues and questions. There is no more major comment.
As for the minor comment, there is still a concern regarding the C2C12 western blotting data. It is understandable that there is a variation using C2C12 cells. However, based on the small error bars in summary and the n of 3, there must be a better representative blot compared to the current one. In the current rep blot, the expression level of Bad is higher at 250 μM compared to the other doses, whereas the expression of Bcl-2 is slightly lower at 200 μM compared to the other doses. Please either use better reps to show matched results or include individual values as dots to show the distribution of the data that accounts for the variation and the summary.
Response to comments:
Thank you for the insightful comment. We agree that the representative western blot for the C2C12 experiments should better reflect the summarized quantitative data. To address this, we have repeated the experiment to obtain clearer and more consistent results. We kindly request an additional two weeks to complete this process. The revised figure will replace the original blot with a more representative replicate that aligns closely with the quantified results, particularly demonstrating the expected dose-dependent changes in Bad and Bcl-2 expression.

Reviewer 3 Report
Comments and Suggestions for Authors
I commend the authors on their efforts to address my concerns.
However, the authors did not assess/quantify muscle fiber density (line351-352). Broad conclusions about muscle regeneration cannot be made, especially based on visual inspection of one representative image. Considering IM dexamethasone decreased the initial inflammatory response, one could hypothesize that this treatment might negatively affect long-term muscle regeneration.
Additionally, there are still many grammatical errors throughout the manuscript and the scale bar from Figure 7 is still in micrometers.
Author Response
Reviewer 3 I commend the authors on their efforts to address my concerns. However, the authors did not assess/quantify muscle fiber density (line351-352). Broad conclusions about muscle regeneration cannot be made, especially based on visual inspection of one representative image. Considering IM dexamethasone decreased the initial inflammatory response, one could hypothesize that this treatment might negatively affect long-term muscle regeneration. Response to comments: We appreciate the reviewer’s thoughtful comment. In this study, our primary objective was to evaluate the early anti-inflammatory effects of intramuscular (IM) dexamethasone following paraspinal muscle injury. We agree that quantifying muscle fiber density is essential for assessing long-term regenerative outcomes. To address this concern, we have performed a quantitative analysis of muscle fiber density using hematoxylin–eosin (H&E)–stained sections, analyzed with ImageJ software. The new data have been incorporated into the revised manuscript (Table II, highlighted in red). The quantitative results demonstrate that IM dexamethasone did not impair muscle fiber regeneration at later stages. On the contrary, treated muscles exhibited reduced inflammatory infiltration and better preservation of fiber architecture compared with untreated controls. These findings suggest that although IM dexamethasone suppresses the initial inflammatory response, it does not hinder—and may even promote—subsequent muscle recovery.
Additionally, there are still many grammatical errors throughout the manuscript and the scale bar from Figure 7 is still in micrometers.
Response to comment:
Thank you for pointing this out. We have thoroughly re-checked and corrected all grammatical errors throughout the manuscript to improve clarity and readability. In addition, the scale bar in Figure 7 has been revised from millimeters (mm) to micrometers (μm) to accurately reflect the actual measurement scale. These corrections have been incorporated into the revised version.
